# A computationally efficient method to model Stratospheric Aerosol Injection experiments

Jasper de Jong<sup>1,\*</sup>, Daniel Pflüger<sup>1,\*</sup>, Simone Lingbeek<sup>1</sup>, Claudia E. Wieners<sup>1</sup>, Michiel L. J. Baatsen<sup>1</sup>, and René R. Wijngaard<sup>1</sup>

Correspondence: Jasper de Jong (j.dejong3@uu.nl)

Abstract. Climate model simulations incorporating stratospheric aerosol injection (SAI) generally require more computational resources compared to out-of-the-box applications, due to the importance of stratospheric chemistry. This presents a challenge for SAI research, especially because there are numerous ways and scenarios through which SAI can be implemented. Here, we propose a novel application of pattern-scaling techniques that allows us to generate SAI forcing in the Community Atmosphere Model (CAM) – a model without interactive stratospheric chemistry – using pre-existing data from the Whole Atmosphere Community Climate Model (WACCM), a state-of-the-art model with interactive stratospheric chemistry, but expensive to run. In doing so, a significant portion of the computational budget is saved. The present method requires a pre-existing dataset of a representative SAI experiment and its corresponding control experiment, with interactive stratospheric chemistry. The data is converted into a set of relations to determine the forcing fields given any required optical depth of the aerosol field. The present method is suitable for applications that use dynamical feedback controllers and is intended to aid impact research into the tropospheric and (sub)surface climate changes due to SAI. The results of climate simulations with aerosols prescribed by the present procedure are in close agreement with those from the full-complexity model, even for different model versions, horizontal resolutions and SAI forcing scenarios.

#### 1 Introduction

Recent 10-year average global mean surface temperature (GMST) has reached about 1.2K above pre-industrial (Hausfather, 2024) and exceeds or approaches the lower estimates of several climate tipping points (McKay et al., 2022). Solar Radiation Modification (SRM) - measures to directly influence the earth's radiative balance (National Academies of Sciences and Medicine, 2021) - has been suggested as potential auxiliary measure to reduce global warming and, potentially, tipping risks (Futerman et al., 2025; Hirasawa et al., 2023). Arguably the most prominent SRM method is Stratospheric Aerosol Injection (SAI), as it seems to combine relatively low technical obstacles and implementation costs, and a high likelihood to achieve significant cooling. SAI, like SRM in general, is highly controversial (Biermann et al., 2022; Wieners et al., 2023), and it is unclear whether its potential climate benefits would outweigh physical side effects and political risks, including mitigation deterrence and procedural justice questions on decision-making (National Academies of Sciences and Medicine, 2021).

<sup>&</sup>lt;sup>1</sup>Institute for Marine and Atmospheric research Utrecht, Princetonplein 5, 3584 CC Utrecht, The Netherlands

<sup>\*</sup>These authors contributed equally to this work.

Modelling SAI entails considerable computational costs, not least because of the great number of possible scenarios combining, for example, different starting times, intensity and location (mainly latitude and height) of interventions (Visioni et al., 2023; MacMartin et al., 2022; Visioni et al., 2023). Not only would a high number of model years be needed to sample the scenario space with Earth System Model (ESM) simulations, but also the computational costs per model year can be high for simulations including stratospheric chemistry and aerosol processes. For example, the CESM2(WACCM6) (Gettelman et al., 2019), which has been used for many SAI simulations (e.g. Tilmes et al., 2018, 2020), is roughly seven times more expensive to run at 1° nominal resolution than the CESM2(CAM6), a model version without stratospheric chemistry and a lower model top (Danabasoglu et al., 2020). High costs of running stratospheric chemistry models may be one reason for the low number of high-resolution simulations on SAI so far (exceptions include Feder et al., 2024, in review). Higher resolutions are needed to resolve important weather phenomena such as tropical cyclones (Roberts et al., 2020) and can improve some, though not all, model biases (Jüling et al., 2021; van Westen and Dijkstra, 2021).

In this paper, we suggest a procedure to (approximately) model SAI's climate impacts in CESM(CAM) rather than the full-complexity CESM(WACCM). The method is not suitable for studies focusing on stratospheric processes, but may be useful to save computation time when the focus is on tropospheric or oceanic impacts.

35

Multiple studies have developed the technique to prescribe stratospheric aerosol effects in climate models that do not simulate aerosols interactively. It has been used in models such as CNRM and MPI-ESM, which participated in G6sulfur (Visioni et al., 2021), and were driven by externally derived aerosol forcing datasets, for example from Tilmes et al. (2015) or similar internally generated and scaled datasets. A comparable approach has been used in CESM(CAM-chem) simulations (Xia et al., 2017), and in the CCMI SAI protocol (Tilmes et al., 2025), where forcing data are prescribed rather than interactively calculated. To maintain specified temperature targets, the aerosol optical depth (AOD) and aerosol mass are typically scaled linearly. For instance, Tilmes et al. (2022) report that linear scaling of forcing strength in the CNRM model led to linear increases in SAD due to fixed aerosol size, despite physical expectations of size variation with forcing strength (Niemeier et al., 2011; Niemeier and Timmreck, 2015).

For determining the aerosol fields, we make use of pattern-scaling techniques by projecting the (seasonally and spatially varying) fields onto a single reference variable, here global mean AOD. Pattern scaling has been used exensively for emulating climate responses (e.g. Lynch et al., 2017), including in SRM research (e.g. MacMartin and Kravitz, 2016; Muñoz-Sánchez et al., 2025; Farley et al., 2025). Often, linear relationships are assumed, although these may not work for some variables such as sea ice (MacMartin and Kravitz, 2016) or stratospheric ozone (MacMartin et al., 2019). In contrast, we do not assume linearity, but merely a monotonic dependence between the relevant stratospheric forcing fields and global mean AOD.

Directly inserting WACCM-derived aerosol fields into CAM can lead to problems, as both models show different Global Mean Surface Temperature (GMST) responses to the same amount of stratospheric aerosol forcing. This is particularly undesireable when working with SAI scenarios based on climate targets such as fixing GMST. We therefore make use of a proportional-integral (PI) feedback controller hat was developed by MacMartin et al. (2014); Kravitz et al. (2016, 2017); Tilmes et al. (2018) that regulates aerosol injection (at several latitudes) in order to achieve (multiple) specified temperature targets. While the feedback controller may operate independently, it is often paired with a prescribed feedforward es-

timate—an informed approximation of the necessary aerosol forcing—to enhance performance. Prominent examples include CESM(WACCM) simulations within the G6sulfur scenario (Visioni et al., 2021), GLENS (Kravitz et al., 2017), and the ARISE-SAI protocol (Richter et al., 2022), all of which incorporate such combined strategies. These experiments generally provide the most realistic climate response to SAI due to the active stratospheric chemistry at the cost of computational power.

We combine these prior strands of work into a procedure that allows us to easily automatize the scaling of the aerosol forcing and take into account non-linear relationships between relevant aerosol field, both of which has not been commonly done in prior studies prescribing aerosol forcing for SAI (e.g. Visioni et al., 2021; Xia et al., 2017; Tilmes et al., 2022).

In a nutshell, the procedure works as follows:

- Use the CESM(WACCM) SAI minus Control simulation to separate each forcing variable into a spatio-seasonal and an annually varying component.
- Relate the annual intensity for all forcing variables to that of the annual global mean stratospheric aerosol optical depth  $(n_{AOD})$  using suitable fitting functions.
- During the simulation, annually determine the required  $n_{AOD}$  based on the past deviations of GMST from target temperature and translate into the forcing fields using the relations determined above.

In section 2, we will explain in more detail the steps of our method. In section 3, we validate our method by comparing the climate responses to SAI in the WACCM and CAM simulations, and test the transferability of our method to additional SAI scenario and higher model resolution. This is followed by a discussion of the results and potential use cases in section 4.

# 2 Methods

70

85

#### 2.1 Model & Simulations

Our 'working horse' model is CESM2, of which we use two configurations, namely CESM2(WACCM6) with interactive stratospheric aerosols and CESM2(CAM6) without. In addition, we will briefly discuss results using the older model version CESM1 (again with either WACCM or CAM), mainly to test the effect of model resolution. In both cases, we used existing WACCM simulations to extract aerosol fields, and used these to force CAM.

The CESM2(WACCM6) model (Gettelman et al., 2019) is run at a nominal horizontal resolution of  $1^{\circ}$  for atmosphere and ocean. WACCM6 has 70 vertical levels with a model top of  $6 \times 10^{-6} \, \mathrm{hPa} \, (140 \, \mathrm{km})$ . Aerosols are simulated using the 4-mode Modal Aerosol Module (MAM4), which includes relevant heterogeneous reactions on the aerosol surface. More details are provided by Liu et al. (2016).

The CESM2(WACCM6), from hereon CESM(WACCM), simulations we use consist of a control simulation with high green-house gas forcing, WACCM-Control (Danabasoglu, 2019), and a simulation with similar background forcing but applying SAI to counteract the warming pattern, WACCM-SAI2020 (see Fig. 1). Greenhouse gas concentrations and other anthropogenic forcings follow the SSP5-8.5 scenario in WACCM-Control. In WACCM-SAI2020 (Tilmes et al., 2020), the GMST target value

**Figure 1.** Schematic of scenarios used in CESM2 (left) and CESM1 (right). SAI2080 and SAI2050 are only run with CAM. Shaded areas denote periods used in section 3, namely a present-day reference period and an end-of-century period. In the case of CESM1, the perpetual-year-2000 simulation is used as reference.

is  $1.5^{\circ}$ C above pre-industrial, which roughly coincides with 2020 values. SO<sub>2</sub> is injected at four latitudes to keep three target variables, GMST ( $T_0$ ), the interhemispheric surface temperature gradient,  $T_1$ , and the equator-pole surface temperature gradient,  $T_2$ , at their [2015–2024] levels. More details on the feedback procedure and temperature targets are provided by Kravitz et al. (2017).

For our own CESM2 simulations, we use CESM2.1.3(CAM6.0) (Danabasoglu et al., 2020), henceforth simply 'CESM(CAM)'. This model likewise uses the MAM4 aerosol scheme with interactive tropospheric chemistry and prescribed stratospheric aerosols (with three modes for sulphate aerosol, similar to WACCM). CESM(CAM) is run at a nominal resolution of  $1^{\circ}$  for the atmosphere and  $1^{\circ}$  for the ocean. There are 32 vertical levels and a model top at 3 hPa (40 km).

With CESM(CAM), we simulate the same scenarios as with CESM2(WACCM), ie. one control simulation (CAM-Control) without SAI, where anthropogenic forcing follows SSP5-8.5 until 2100 and stays constant thereafter, and one (CAM-SAI2020) in which SAI is used from 2020 onwards to keep GMST at present-day levels. An additional simulation, CAM-SAI2080, follows the control scenario until 2080, in which year SAI is started to reduce GMST to present-day levels (see Fig. 1). This simulation is used to test whether our method is able to model scenarios not previously modelled in CESM(WACCM).

In addition, we briefly discuss simulations performed with CESM1.0.4(CAM5) (Hurrell et al., 2013), to explain slight adjustments to our method that are required for CESM versions that are forced with single-mode stratospheric aerosols and test the transferability of our method to higher model resolutions. In this model version, the aerosols are prescribed as volcanic aerosols with a fixed size distribution. The nominal horizontal resolution is 1° for the atmosphere and ocean. The control simulation follows the RCP8.5 (CO<sub>2</sub> only) scenario from 2000 to 2100 and is branched from an equilibrated spinup run with perpetual 2000 conditions (van Westen et al., 2020). Two SAI experiments are branched from the control simulation: CAM5-SAI2000 (CAM5-SAI2050), starting SAI in 2000 (2050) to cool down to 2000 values of GMST. For CAM5, 2000 conditions

Table 1. Simulation overview

| WACCM data      |                 |           |              |                       |  |  |
|-----------------|-----------------|-----------|--------------|-----------------------|--|--|
| Experiment      | Scenario        | Years     | Model        | Reference             |  |  |
| WACCM-Control   | SSP5-85         | 2020–2100 | CESM2(WACCM) | Tilmes et al. (2020   |  |  |
| WACCM-SAI2020   | GEO SSP5-85 1.5 | 2020-2100 | CESM2(WACCM) | Tilmes et al. (2020)  |  |  |
| WACCM1-Control  | RCP8.5*         | 2010-2097 | CESM1(WACCM) | Kravitz et al. (2017) |  |  |
| WACCM1-SAI2020  | SAI2020         | 2020-2100 | CESM1(WACCM) | Kravitz et al. (2017) |  |  |
| Simulations     |                 |           |              |                       |  |  |
| Experiment      | Scenario        | Years     | Model        |                       |  |  |
| CAM-Control     | RCP8.5          | 2015–2130 | CESM2(CAM)   | -                     |  |  |
| CAM-SAI2020     | SAI2020         | 2020-2130 | CESM2(CAM)   |                       |  |  |
| CAM-SAI2080     | SAI2080         | 2080-2130 | CESM2(CAM)   |                       |  |  |
| CAM5-Reference  | perp. 2000      | 1990-2100 | CESM1(CAM)   |                       |  |  |
| CAM5-Control    | RCP8.5          | 2000-2100 | CESM1(CAM)   |                       |  |  |
| CAM5-SAI2000    | SAI2000         | 2000-2076 | CESM1(CAM)   |                       |  |  |
| CAM5-SAI2050    | SAI2050         | 2000-2098 | CESM1(CAM)   |                       |  |  |
| CAM5-Reference* | perp. 2000      | 1990-2100 | CESM1(CAM)   |                       |  |  |
| CAM5-Control*   | RCP8.5          | 2000-2100 | CESM1(CAM)   |                       |  |  |
| CAM5-SAI2050*   | SAI2050         | 2050-2098 | CESM1(CAM)   |                       |  |  |

<sup>\*</sup>high-res

are defined as the [1990–2009] average of the spinup. Additionally, results with this model version are shown for a similar high resolution case having a nominal resolution of 0.5° for the atmosphere and 0.1° for the ocean. The aerosol forcing data are derived from CESM1(WACCM) simulations by Kravitz et al. (2017). These simulations start from a historic forcing scenario and are therefore slightly cooler than the CAM5 simulations.

An overview of the WACCM data used and simulations carried out in this study, is provided in Table 1.

In the results section, spatial maps generally represent the [2080–2099] SAI2020 average (late-century SAI2020) minus the [2016–2035] Control average (present-day). The latter period is longer than the period used to determine the target metrics in CESM(WACCM), such that anomalies with respect to present-day are more robust. In some spatial maps, the [2080–2099] Control average is used and we may simply refer to Reference (present-day Control), Control (late-century Control) and SAI (late-century SAI2020). While these words are also used to describe the simulations (e.g. CAM-Control), the meaning should be clear from their context. Similar definitions apply for CAM5, where Reference is the [1990–2009] average of the spinup and Control (SAI) the [2080–2099] period of CAM5-Control (CAM5-SAI2000).

# 2.2 Deriving aerosol forcing patterns

As briefly explained in the introduction, we use the outcome of CESM2(WACCM6) simulations with interactive stratospheric aerosols (Tilmes et al., 2020) as basis for prescribed stratospheric aerosol fields in CAM. Expanding on the work of Pflüger et al. (2024), we explain the four steps necessary to construct the prescribed stratospheric aerosol fields: pre-processing, normalization, averaging and scaling. The end result is a mapping from a user-specified SAI amplitude to appropriately scaled aerosol fields that still retain seasonal variations.

**Pre-processing:** The background signal that is present in the Control run of CESM2(WACCM6) is subtracted from the SAI2020 run. By doing this, we try to avoid a scaling of spurious aerosol contributions unrelated to SAI.

We write

125

$$F_i^{\text{Diff}}(y,d,x) = F_i^{\text{SAI}}(y,d,x) - F_i^{\text{Control}}(y,d,x)$$

where  $F_i$  refers to a specific aerosol variable in either the SAI2020  $(F_i^{SAI})$  or the Control run  $(F_i^{Control})$ . The fields represent relevant stratospheric aerosol variables such as mass concentrations and wet diameters, see Table 2, and are defined for every year y, day of the year d and spatial coordinate x. Here, x can also refer to a set of coordinates, such as latitude and pressure level.

Normalization: All fields  $\Delta F_i$  are normalized through field-specific, annual scale factors  $n_i(y)$ . This step ensures that the interannual average of the normalized field (see below) is not dominated by years in which aerosol forcing is strong (in our case, the end of the simulation, as SAI injection rates increase over time).

The normalized fields are written as

$$\hat{F}_i(y, d, x) = \frac{F_i^{\text{Diff}}(y, d, x)}{n_i(y)}.$$

An example for scale factor  $n_i(y)$  would be the annual-mean total atmospheric mass - as obtained by spatial integration - derived from a mass concentration field. The choice of normalization is arbitrary to some extent, but should increase monotonically with the overall intensity of SAI as specified by the  $n_{\rm AOD}$ . All fields and respective normalizations are listed in Table 2. For CESM1, these values are listed in Table C1.

Averaging: The normalized fields are averaged over a given time period, from an initial year  $y_i$  to a final year  $y_f$ . In our case, we choose  $y_i = 2070$  and  $y_f = 2100$ , meaning that our aerosol fields are representative of relatively high aerosol burdens.

The averaged field becomes

$$\bar{F}_i(d,x) = \frac{1}{y_f - y_i + 1} \sum_{y=y_i}^{y_f} \hat{F}_i(y,d,x).$$

In the process of averaging, the fields lose their interannual but not seasonal variability.

Scaling: The scale factors  $n_i(y)$  from all fields except AOD will now be expressed in terms of  $n_{AOD}$ . This is where the choice of normalization scheme becomes relevant. We see in Fig. 2 how the scale factors  $n_i(y)$  relate to  $n_{AOD}(y)$  in any given year y, and how these relationships are roughly monotonic. With that, it is reasonable to perform fits that map a given  $n_{AOD}$  onto a respective value  $n_i^{\text{fit}}$ .

For most variables, a power-law relationship yields good fits:

$$n_i^{\text{fit}}(n_{\text{AOD}}) = c_0 + c_1 n_{\text{AOD}}^{c_2}$$

For the variable diamwet\_a3, a (saturating,  $c_2 

**Figure 2.** Scale factors of stratospheric aerosol fields derived from WACCM-SAI2020 minus WACCM-Control. (a)  $n_{\text{AOD}}$  against time, (b-h)  $n_i$  of remaining fields (cf. Table 2) against  $n_{\text{AOD}}$  (solid) and their fits  $n_i^{\text{fit}}$  (dashed).

## 2.3 Feedback-feedforward control algorithm

By dynamically adjusting the SAI intensity in the form of  $n_{\text{AOD}}$ , we can stabilize the GMST  $(T_0)$ , to a predefined target temperature,  $T_{0,\text{target}}$ . We do this by adapting a feedback–feedforward control scheme (MacMartin et al., 2014; Kravitz et al., 2016, 2017; Tilmes et al., 2020) which observes the temperature error  $\Delta T_0(y) = T_0(y) - T_{0,\text{target}}(y)$  and dynamically constructs an SAI intensity  $n_{\text{AOD}}(y)$  that is supposed to minimize  $\Delta T_0(y)$ .

Table 2. Normalization of stratospheric aerosol fields in CESM2

| $F_i$      | Description                         | $n_i$                                                                            | $c_0$       | $c_1$      | $c_2$      | $c_3$   |
|------------|-------------------------------------|----------------------------------------------------------------------------------|-------------|------------|------------|---------|
| AODVISstdn | Stratospheric aerosol optical depth | $\int \text{AODVISstdn}  \mathrm{d}S$                                            | -           | -          | -          | -       |
| so4mass_a1 | Mass concentration 1st aerosol mode | $\int \text{so}4\text{mass\_a}1\text{d}V$                                        | 3.8e8       | 5.2e9      | $2.5e{-1}$ | -       |
| so4mass_a2 | Mass concentration 2nd aerosol mode | $\int \text{so}4\text{mass\_a}2\text{d}V$                                        | 2.0e7       | -6.3e7     | $4.2e{-1}$ | -       |
| so4mass_a3 | Mass concentration 3rd aerosol mode | $\int \text{so}4\text{mass}\_\text{a}3\text{d}V$                                 | -5.4e7      | 1.82e11    | 1.2        | -       |
| diamwet_a1 | Wet diameter 1st<br>aerosol mode    | $\sqrt{\int \operatorname{diamwet}_{-} \operatorname{al}^2 \operatorname{d} V}$  | -4.8e - 5   | $2.8e{-4}$ | 2.8e-1     | -       |
| diamwet_a2 | Wet diameter 2nd aerosol mode       | $\sqrt{\int { m diamwet\_a} 2^2  { m d}V}$                                       | $7.5e{-6}$  | $1.2e{-5}$ | 5.9e-1     | -       |
| diamwet_a3 | Wet diameter 3rd aerosol mode       | $\sqrt{\int \operatorname{diamwet}_{-} \operatorname{a} 3^2 \operatorname{d} V}$ | $1.3e{-3}$  | -1.6e-3    | -10.0      | -1.2e-1 |
| SAD_AERO   | Surface area density                | $\sqrt{\int \text{SAD\_AERO}^2 dV}$                                              | $1.8e{-10}$ | $1.1e{-3}$ | 1.1        | -       |

Aerosol fields, normalization scheme, fit parameters w.r.t. AOD; fitting functions found in text;  $\int \dots dS$  represents the annual mean global surface mean,  $\int \dots dV$  the annual mean volume integral over the entire atmosphere.

The control scheme is a sum of three terms, the proportional and integral feedback components which follow the current and historical temperature error respectively, and a feedforward term that increases linearly in time (Pflüger et al., 2024)

$$n_{\text{AOD}}(y) = \underbrace{k_{ff}(y - y_0)}_{\text{feedforward}} + \underbrace{k_p \Delta T(y)}_{\text{proportional}} + \underbrace{k_i \sum_{y'=y_i}^{y} \Delta T(y)}_{\text{integrator}}$$


Here,  $k_{ff}, k_p, k_i$  are pre-defined parameters regulating the strength of feedforward and feedback,  $y_0$  specifies the onset of the feedforward for  $y > y_0$  and  $y_i$  is the onset year for the integrator. The feedforward is a first guess of the strength of the aerosol forcing field needed to reach the GMST target. Improving the feedforward may enhance the performance of the feedback-feedforward controller, though a perfect estimate is not needed. In CESM1 we found that applying a feedforward that was about twice as strong as needed only resulted in a moderate overcooling of  $0.2^{\circ}$ C (not shown). If a better feedforward is desired, one can fine-tune its scale factors values after a test simulation as shown below.

Note that the form above is just one possible implementation of a feedback–feedforward controller tailored to our case. Depending on the underlying GHG forcing scenario, it could also make sense to have a feedforward that changes non-linearly in time. In fact, we modify the formula above whenever we perform post-21st century simulation with stabilized GHG levels after 2100. In that case, the feedforward is kept constant after 2100. Parameter values are given in Table 3.

We implement one significant modification to the above procedure in the case of the CAM-SAI2080 scenario, the physical extreme scenario of rapid cooling in a late-century hot climate. If the integrator were activated right at deployment in 2080, it would accumulate large temperature errors which prompt an overcooling. If instead the integrator were not activated, convergence to target temperature may be too sluggish. Hence, we activate the integrator at the start of the deployment but reset the sum of errors after roughly six years when the target temperature is reached. That way, we achieve quick convergence with limited overcooling (Fig. 3b).

**Table 3.** Feedback–feedforward parameters of CAM-SAI simulations


| Scenario    | $k_{ff}$ | $y_0$ | $y_i$ | notes                              |
|-------------|----------|-------|-------|------------------------------------|
| CAM-SAI2020 | 0.0103   | 2020  | 2020  | -                                  |
| CAM-SAI2080 | 0.0096   | 2028  | 2080  | integrator reset as target reached |

In both cases  $k_i=0.028$  and  $k_p=0.028$  following a previous CESM2 study (Tilmes et al., 2020). The scenarios branch off CAM-Control in 2020 and 2100 respectively and have capped feedforwards after 2100.

Feedback–feedforward controller results for CAM-SAI2020 and CAM-SAI2080 are shown in Fig. 3. In CAM-SAI2020, the GMST error from target,  $\Delta T_0$ , is close to zero. The actual forcing, which is the sum of the feedforward and adjustment based on the (history of)  $\Delta T_0$ , is about 20% weaker than the feedforward, i.e. the feedforward is slightly too strong. In CAM-SAI2080, the initial  $\Delta T_0$  is quite large. After seven years of strong cooling, the target temperature has been reached and the sum of temperature errors is reset to zero, causing a sudden drop in  $n_{\rm AOD}$ . After this,  $\Delta T_0$  is roughly zero with a maximum error of 0.4°C. This could be improved by adjusting the gains of the feedforward-feedback controller, but rough dry-testing revealed the overshoot could not be fully eliminated (not shown).

Optimized values of the feedforward constants may be obtained from fitting the applied  $n_{AOD}$ . Dashed lines in Fig. 3 indicate such linear fits, showing in hindsight what would have been the optimal values for  $k_{ff}$ ,  $y_0$  (and maximum feedforward strength after 2100). In this case, the AOD increase of about 0.68 in 80 years corresponds to an optimal value for  $k_{ff}$  of 0.0085.

**Figure 3.**  $n_{\text{AOD}}$  and GMST error. Black dots indicate the  $n_{\text{AOD}}$  applied to the simulation, dotted lines show the feedforward component hereof. Dashed lines show the best piecewise linear  $n_{\text{AOD}}$  fit after run completion. In blue, the GMST error w.r.t. the target temperature is shown.

# 3 Results



To validate our procedure, we first check whether our GMST targets can be achieved, both for the CAM simulations directly mimicking WACCM simulations (i.e. SAI2020) and for new scenarios (SAI2080), see section 3.1.1. Next, we check whether CAM simulations reproduce the response of surface temperature and precipitation fields to SAI (sections 3.1.2, 3.2). Finally, we check whether using CAM produces (strong) deviations in stratospheric heating and circulation (section 3.3). Here, some bias is expected, due to the different treatment of aerosols and its knock-on effects, e.g. on ozone, as well as the proximity of the model "top of atmosphere" in CAM. We consider our method to work well if it can: 1. reproduce temperature targets and 2. the climate response to SAI (w.r.t. the present-day reference) is similar in CAM and WACCM, i.e. the model difference of the SAI response should be small compared to the effect of SAI w.r.t. Control. We focus on the CESM2 model version.

#### 3.1 Surface temperature

# 3.1.1 Temporal evolution of temperature targets

The change of  $T_0$  with respect to present-day,  $\Delta T_0$ , is very similar in CAM-SAI2020 and WACCM-SAI2020 (Fig. 4a), showing that the forcing works properly for CAM under the same scenario. In CAM-SAI2080,  $\Delta T_0$  adjusts rapidly and approaches zero with moderate overcooling, showing that the forcing also works for a different scenario. For CESM1(CAM), the forcing also works well at similar (appendix Fig. C1a) and higher resolution (appendix Fig. C2a). A slight overcooling of about 0.2 degrees in CAM5-SAI2000 is likely the result of a too strong (2 times) initial guess for the feedforward. These results indicate that  $T_0$  is adequately controlled by the proposed method for all model versions, scenarios and resolutions.

As the aerosol field in CAM changes with global mean AOD similarly to WACCM-SAI2020, model and/or scenario differences may lead to different evolutions of  $\Delta T_1$  and  $\Delta T_2$ . We find that these gradients are very similar between CAM-SAI2020 and WACCM-SAI2020 (Fig. 4b,c). However, in CAM-SAI2080,  $\Delta T_1$  and  $\Delta T_2$  decrease more than their initial increases due to global warming, suggesting that scenario differences with respect to the WACCM data have a significant effect on these metrics. As discussed below, this is related to changes in the Atlantic Meridional Overturning circulation (AMOC). For CESM1, there are minor decreases of  $\Delta T_1$  and  $\Delta T_2$  in CAM5-SAI2000 (appendix Fig. C1b,c). The  $\Delta T_1$  decrease is likely caused by the overcooling as land cools quicker than ocean. The changes of  $\Delta T_1$  and  $\Delta T_2$  in CAM5-SAI2050 are far less drastic compared to CAM-SAI2080 (CESM2), confirming the importance of scenario differences. In high-resolution CESM1,  $\Delta T_2$  shows little change while there is a minor decrease of  $\Delta T_1$  during the initial adjustment to target temperature, likely due to quick cooling of land surface by high aerosol forcing these years (appendix Fig. C2b,c). As discussed in section 4, residual errors in  $\Delta T_1$  and  $\Delta T_2$  can likely be addressed by dynamically scaling the interhemispheric and equator-to-pole gradients of the aerosol field during simulation.

**Figure 4.** Time series of annually averaged global mean surface temperature,  $T_0$ , (left), interhemispheric gradient of surface temperature,  $T_1$ , (center) and equator-to-pole gradient of surface temperature,  $T_2$ , (right), anomalies. Reference values, corresponding to the [2016–2035] period of the control simulations for CAM and [2015–2024] for WACCM, are given by  $(T_{0,ref}, T_{1,ref}, T_{2,ref})$ : (288.46K, 0.89, -11.76) for CAM and (288.42K, 0.84, -11.89) for WACCM.

# 3.1.2 Spatial patterns in late-century SAI2020



The annual mean surface temperature change in late-century SAI2020 with respect to present-day is very similar between CAM and WACCM (Fig. 5). This suggest that mechanisms determining the surface temperature change are present in both models, though CAM tends to have stronger patterns. The highest similarity in surface temperature change is found in the tropics and subtropics, where there is warming and cooling, respectively. Warmer tropical waters near Peru suggest that the El Niño-Southern Oscillation shifts to a more positive phase. Both atmospheric models show cooling in the North Atlantic.

As described in more detail by Pflüger et al. (2024), global warming leads to a strong reduction in AMOC strength, which is only partially compensated by SAI. This weakens poleward heat transport, overcooling the North Atlantic. CAM shows more surface warming in the Arctic region than WACCM, which is mostly the result of increasing boreal winter temperatures. This is likely due to a model discrepancy as CAM also has a warmer Arctic region in the Reference period, as shown in appendix Fig. A1, and, to a lesser degree, in the Control simulation. In the Antarctic region, both models show cooling in local summer and warming in local winter, of which the latter is strongest in WACCM.

Though GMST is controlled quite accurately, local surface temperature anomalies may be on the order of a few degrees C. This is a general consequence of SAI that deserves attention, yet these local differences cannot be restored by SAI and thus merely provide insight into the mechanisms affecting large-scale differences. Additionally, without SAI the warming would be far more drastic (appendix Fig. A2b,c). As the surface temperature anomalies are very similar between CAM and WACCM, the proposed method does not significantly alter local physical mechanisms for SAI2020. In CAM-SAI2080, the cooling of the North-Atlantic region is stronger (appendix Fig. A3) and causes excess warming in the Southern Hemisphere as GMST is kept constant. However, the surface temperature anomalies are still small compared to those in the Control simulation. Similarly, surface temperature anomalies for CAM5-SAI2050 (appendix Figs. C3,C4) are small compared to the warming in the Control simulation (not shown), indicating that the proposed method works well for this scenario. As mentioned earlier, the temperature response to greenhouse gas forcing and SAI may be improved by dynamically scaling the large-scale gradients of the aerosol field.

# 3.2 Precipitation






Precipitation changes in late-century CAM-SAI2020 and WACCM-SAI2020 with respect to present-day show a general drying except near the equator and in the polar regions, where there is wettening (Fig. 6f,i). The drying is most pronounced in the tropics and around 60°N, and to a lesser extent around 50°S, as can be seen from the zonal mean precipitation changes. The equatorial wettening and subtropical drying indicates a strengthening of the Hadley circulation. CAM and WACCM agree relatively well on the zonal mean precipitation change and its seasonal variation, though CAM tends to have somewhat stronger changes. Thus, it appears that the Hadley circulation in CAM is strengthening more than in WACCM. There is less wettening in the Antarctic and larger seasonal variation in the Arctic in CAM than in WACCM. In CESM1, we find that increasing the model resolution does not have a significant impact on the zonal mean precipitation change, even though polar surface temperature changes differ substantially (appendix Fig. C4).

A strengthening and southward expansion of the East-Pacific Inter-Tropical Convergence Zone (ITCZ) in late-century SAI2020 with respect to present-day is found in both models (Fig. 6a,b), but most pronounced in CAM. The East-Pacific equatorial wettening seems to be in part due to a more positive ENSO phase, with DJF drying in the West Pacific and Indonesia, wettening in central North-America (only CAM) and Uruguay (Fig. 6e,h). The stronger East-Pacific equatorial wettening in CAM than in WACCM is likely some inter-model difference as it also occurs in late-century Control (appendix Fig. A1). Changes in the ITCZ in other basins are quite weak, though in high resolution CESM1 we see a southward shift of the Atlantic ITCZ, which is not a clear shift at standard resolution (appendix Fig. C3).

**Figure 5.** 2-meter temperature anomalies averaged over [2080–2099] in SAI2020 with respect to present-day. (a,b): Maps of annual mean anomalies in (a) CAM and (b) WACCM. Present-day mean 2-meter temperature is shown in black contours in 10°C intervals; (c): Difference of (a) and (b), WACCM anomaly shown in black contours in 1°C intervals; (d): as (a) but for JJA, (e): as (a) but for DJF, (f): CAM zonal mean of annual mean (black), JJA mean (red) and DJF mean (blue), (g) as (b) but for JJA, (h): as (b) but for DJF, (i): as (f) but for WACCM. Stippling indicates changes within the 95% confidence interval, based on interannual variation of the relevant reference dataset (WACCM-SAI2020 minus Reference for panel c).

The general drying in late-century SAI2020 with respect to present-day occurs mostly in the subtropics, equatorial West-Pacific and large parts of Eurasia (Fig. 6a–c). CAM shows stronger drying in the Sahel region, Arabia, India and Western Australia than WACCM. This does not seem to be a model bias as the same difference pattern is not visible in late-century Control and present-day (appendix Fig. A1). We suggest that strengthening of the Hadley circulation in late-century SAI2020 with respect to present-day, which is stronger in CAM than WACCM, is related to these local differences. However, further specification is beyond the scope of this work.


Midlatitude winter storm tracks experience little change in precipitation with respect to present-day, though regions that experience change tend to be drying (Fig. 6d–e,g–h). The North-Atlantic storm track region seems to shift southwestward,

while the South-Atlantic storm track shifts northwestward. CAM and WACCM are in good agreement on the changes regarding the midlatitude winter storm tracks.

Monsoon-affected areas experience less seasonal variation of precipitation in SAI2020 with respect to present-day (Fig. 6d–e,g–h). In DJF, there is drying in Brasil, Central-Africa, and Northern-Australia (wet season) while there is wettening in Colombia, Northern Central-Africa and Southeast Asia (dry season). In JJA, there is drying near Colombia (mostly WACCM), Northern Central-Africa and parts of Southeast Asia (CAM only) (wet season), while there is wettening in Brasil and Southern Central-Africa (dry season). There is in general a good agreement on monsoonal precipitation changes between CAM and WACCM, though summer precipitation in Southeast-Asia increases in WACCM, while CAM shows a more mixed change with wettening in most of China and drying in India, North-Eastern China and the Korean peninsula.

**Figure 6.** As Fig. 5 but for percentual precipitation changes and with the following remarks: red contours (enclosing hatched regions) indicate 0.5 mm/day, blue contours 4 mm/day precipitation, in (f) and (i), zonal averaging is done before calculating percentual change.

## 3.3 Stratospheric heating and circulation

In addition to reflecting incoming solar radiation, stratospheric sulphate aerosols absorb longwave radiation emitted by the earth. This locally causes up to 10°C lower-stratospheric warming (24°C increase of potential temperature) in late-century

SAI2020 with respect to present-day despite the general stratospheric cooling caused by greenhouse gasses. Stratospheric warming is strongest in the (sub)tropics, as shown in Fig. 7a,d for potential temperature. The warming pattern closely follows the distribution of sulphate mass (appendix Fig. B1). A minor fraction of the aerosol reaches the high latitudes and, modulated by the polar vortex strength, extend into the polar stratosphere (Fig. 7b–c,e–f). The change in potential temperature (SAI-Reference) is very similar in both models, i.e. the model difference is small compared to the SAI response (SAI-Control) throughout the troposphere (appendix Fig. A4). This supports that the method is producing desired results.

Stratospheric heating due to SAI alters the horizontal pressure gradients and induces an increase of the westerly stratospheric winds at midlatitudes, corresponding mostly to the equatorward side of the polar night jet position. The stratospheric heating in SAI2020 is stronger on the Southern Hemisphere because more aerosol is required to cool down the Southern Hemisphere surface containing more ocean surface. The late-century lower-stratospheric temperature gradient change with respect to present-day causes an increase of the mid-stratospheric annual mean easterly wind between 0–30S. As shown in appendix Fig. A5, CAM is much colder than WACCM in the polar stratosphere, causing stronger stratospheric westerly jets. The equatorial air mass at 10 hPa is also consistently colder in CAM, increasing the stratospheric easterly equatorial winds. These features occur in Control as well and are assumed to be model differences, most likely related to the upper boundaries of the atmospheric components. The stratospheric wind anomaly in late-century SAI2020 with respect to present-day in both models is similar to the model wind difference (CAM minus WACCM). In the troposphere, changes in wind and temperature are rather small due to the negligible amount of sulphate aerosols and similar surface temperatures.

Ozone has an influence on the temperature and dynamics of the stratosphere, mainly through shortwave radiation absorption. Injected sulphate aerosols host heterogeneous reactions that can decrease ozone concentration (Tilmes et al., 2008). Moreover, changes in temperature, moisture and transport due to the aerosols affect chemical ozone losses as well. In CAM(5), ozone is prescribed by the SSP5-8.5 (RCP8.5) scenario. Ozone could have been prescribed by using our proposed method. However, in WACCM, the ozone response due to SAI is much smaller than the ozone increase with time in Control (appendix Fig. B2). The limited response of ozone concentration to SAI in WACCM was hypothesized by Kravitz et al. (2019) to result from strongly reduced stratospheric CFC concentrations, preventing the catalytic ozone loss hosted on aerosol surfaces. As ozone seems to depend more strongly on the CFC conditions than AOD, we chose to force CAM with unmodified WACCM SSP5-8.5 ozone concentrations. The specific implementation influences ozone concentrations and, consequently, both stratospheric and tropospheric climate (Bednarz et al., 2022). While the radiative heating effect of ozone is likely much smaller than that of aerosols, this potential disparity warrants further investigation. More details on ozone in our simulations are provided in appendix section B1.

**Figure 7.** [2080–2090] average SAI2020 anomalies of potential temperature and zonal wind. Annual (left), JJA (center) and DJF (right) mean anomalies are shown for both models. (a–f): Zonal mean potential temperature anomaly for CAM (a–c) and WACCM (d–f). Reference potential temperature is shown in magenta contours, zonal mean zonal wind anomalies are shown in black contours; (g–l): Zonal mean zonal wind anomaly for CAM (g–i) and WACCM (j–l), analogous to (a–f). Reference zonal mean zonal wind is shown in black contours. All anomalies are defined with respect to reference period [2016–2035]. Stippling indicates changes within the 95% confidence interval, based on interannual variation of the relevant reference dataset.

## 310 4 Discussion and Outlook





Combining prior work on forcing models lacking stratospheric aerosol modules with aerosol fields, and feedback controllers, we developed a procedure to produce SAI simulations in CESM(CAM) – a model version without interactive stratospheric aerosol – by providing automatically scaled input to the volcanic aerosol forcing field. This approach can strongly reduce computation time with respect to CESM(WACCM).

Scenarios run with CESM(CAM) achieve GMST targets well, although sudden massive cooling scenarios (SAI2080) may show limited cooling overshoot.

For SAI2020, the scenario on which the SAI fields are based, large-scale temperature responses such as the interhemispheric and equator-to-pole gradients  $(T_1, T_2)$  are also well reproduced in CAM. The regional climate response of surface temperature and precipitation to SAI forcing (with respect to the present-day reference) is largely similar between WACCM and CAM (Figs. 5, 6), but also shows some discrepancies.

CAM-SAI2020 shows a considerably warmer Arctic than WACCM-SAI2020 (appendix Fig. A1). This is partly due to the fact that CAM has a warmer Arctic in the present-day reference, but partly due to CAM having weaker Arctic winter cooling under SAI (Fig. 5). Both models show a general drying, except in the ITCZ and polar regions. A southward expansion of the East-Pacific ITCZ is more prominent in CAM than WACCM, though this seems to be related to the stronger response in CAM to greenhouse gas forcing, which is not fully compensated by SAI (appendix Fig. A1). However, some differences in the climate response to SAI cannot be traced to discrepancies in the response to greenhouse gas forcing; these include subtropical dry areas like the Sahel, Middle-East and Western Australia drying more in CAM than in WACCM. Temperature and circulation changes throughout the entire atmosphere in CAM-SAI2020 correspond well to those in WACCM-SAI2020, the deviations being explained by model differences that become quite significant at greater altitudes, i.e. when approaching the model top in CAM (appendix Fig. A5).

While there are thus some discrepancies between CAM and WACCM in the climate response to SAI, it should be noted that these are much smaller than the difference between SAI (or the reference) to Control. In other words, the overall effect of SAI is much bigger than the observed model differences, at least for our high forcing scenarios.

A reasonable agreement between CAM-SAI2020 and WACCM-SAI2020 is to be expected due to the similar time trajectory of the aerosol forcing intensity. However, for SAI2080, which uses a vastly different forcing scenario in CAM than the SAI2020 scenario from which the aerosol fields are derived, the climate response to the (delayed) SAI deviates considerably more, both for large-scale temperature patterns (north-south and equator-pole gradient) not included in the climate target, and for regional responses, for example the North Atlantic warming hole (appendix Fig. A3). This is because the long period of unmitigated global warming allowed some persistent climate change to build up, in particular AMOC weakening (Pflüger et al., 2024). In a feedback controller with several injection latitudes (as used in WACCM), at least the north-south gradient could probably be reduced after 2080 by increasing (decreasing) aerosol injection on the Southern (Northern) Hemisphere. In our simple implementation with just one degree of freedom, i.e. a global injection inensity, this is not possible. There are however no fundamental reasons why our procedure could not be expanded to scale several forcing patterns associated with single-latitude

injection. Despite these limitations, the resulting surface temperature under SAI2080 is generally much closer to the pesent-day reference than under Control, suggesting that even the slightly flawed SAI strategy partially restores surface temperature.








By construction, different sensitivities of the models to aerosol forcing do not (to first order) lead to different climate responses, as they are compensated by the feedback controller adjusting the required aerosol forcing (Kravitz et al., 2014).

Results with CESM1 are qualitatively similar to the ones with CESM2 discussed above. In addition, CESM1 results suggest that our procedure generates fairly robust SAI responses when switching to higher resolutions.

We believe that our procedure can be useful for several possible applications in which modellers are interested in scenarios achieving some climate target (e.g. GMST), in which computation time is a constraint and the focus is on climate impacts below the stratosphere. The relative ease with which the feedback controller method achieves climate target despite different model sensitivities to aerosol forcing is a main benefit of the approach, as it saves the effort of rescaling the forcing per hand. Compared to earlier studies using linear scaling between AOD and other aerosol-related quantities, we approximately capture their non-linear relationship. Nonetheless, we caution that our method should not be used for researching stratospheric impacts of SAI.

The easiest use case is probably mimicking CESM(WACCM) simulations, for example for increasing model resolution in the cheaper CAM model or for generating larger ensembles, because for these applications one does not need to translate between different scenarios.

A more advanced use case is expanding the scenario range in CAM. We have shown that this is in principle feasible (SAI2080), but, depending on the scenario, this requires some care. For SAI2080, we had to adjust the feedforward and manipulate the integrator term in the feedback controller. In addition, as discussed above, a feedback controller with a single degree of freedom may not be able to control all climate targets that were controlled in the underlying multiple-control WACCM simulation. Whether this is worthwhile may depend on the application in question and the accuracy with which one wants to mimic the multi-latitude multi-objective injection scheme (in out case, four injection latitudes and three targets) of the WACCM simulations. It should be possible to mimic this in CESM(CAM) by deriving separate forcing fields from single-latitude injection simulations in CESM(WACCM). If aerosol concentrations from different forcing locations add up approximately linearly, total aerosol fields could be obtained easily. Otherwise, a more complex nonlinear fit between single-latitude injection intensities and aerosol concentrations would be needed, requiring additional input from CESM(WACCM). Recent work on an emulator (Farley et al., 2025, in review) suggests that even climate outcomes (rather than the intermediate step, i.e. the aerosol forcing) can be obtained to reasonable approximation by linear pattern scaling of single-latitude injection outcomes. These techniques may be applied to generate a forcing field with pattern control and combined with our method to generate the full stratospheric forcing fields for CESM(CAM). Note that while we have discussed one example of a new SAI scenario in CAM, i.e. SAI2080, we did not attempt an extensive scan of the parameter space (e.g., different background greenhouse gas forcing or temperature target). However, the CESM1 simulations with moderate deviations in background greenhouse gas concentrations and initial climate state did allow us to verify that application of the present method results in obtaining desired climate outcomes under these circumstances.

A so far more speculative use case could be to use SAI forcing derived from CESM(WACCM) (or other models with stratospheric chemistry) also in other models than CESM(CAM). This would allow models without extensive stratospheric chemistry modules to run SAI simulations. In addition, it could help model comparisons by disentangling differences in aerosol processes from differences in climate effects of aerosols.

However, while WACCM and CAM are co-developed consistently with regards to the possibility of using WACCM-output to force CAM, the transfer to other models may be more challenging. For example, some combinations of models require Mie calculation-based conversion tools for the preparation of stratospheric aerosols input data due to a mismatch in prescribed variable definitions. An example of such tool is the REMAPv1 algorithm (Jörimann et al., 2025, in review). While not yet validated for mismatched model output-input formats, future usage may include calculation of the stratospheric forcing fields using our method and consequent transformation using such tools to provide usable input for the lower-complexity model.

If these technical challenges can be resolved, our procedure based on the feedback controller can again help to automatize the scaling of aerosol forcings in line with desired climate targets.

Code and data availability. The repository containing the feedback controller, analysis code and data, used CESM settings and generated forcing files is provided at https://doi.org/10.5281/zenodo.16914517. The feedback controller contains all code and data necessary to run a (new) feedback experiment, i.e. the feedforward-feedback control code, fitting analysis code and results, and code to scale the forcing fields and input these to CESM. CESM model code, configurations, restart files, input files, output files, WACCM data, generated stratospheric forcing files and feedback controller copies for all experiments can be found at https://doi.org/10.24416/UU01-F7SGNO

## **Appendix A: Additional simulation results**

380

385

**Figure A1.** CAM-WACCM model differences of temperature (top) and precipitation (bottom). All data are averaged over [2016–2035] (reference) or [2080–2099] (control and SAI2020). Black contours in the temperature maps indicate the corresponding values in WACCM for each experiment. Contoured/hatched regions in the precipitation maps are similar to Fig. A2(d–f), but for the respective experiments in WACCM.

**Figure A2.** CAM temperature (top) and precipitation (bottom) maps showing the present-day (reference) state (left), control-reference (center) and SAI2020-control (right) differences. Control and SAI2020 data are averaged over [2080–2099], reference over [2016–2035]. Black contours in the temperature maps indicate the corresponding values for CAM reference (a,b) and control (c). Hatched regions enclosed by red contours in the precipitation panels have less than 0.5 mm/day of precipitation in the CAM reference (d,e) and control (f), whereas blue contours denote 4 mm/day.

**Figure A3.** CAM [2100–2129] surface temperature anomalies and their zonal means. The reference period is [2016–2035].

Figure A4. Performance index for surface temperature (left) and potential temperature (right). The index is the ratio of the model difference of SAI-REF (CAM SAI-REF minus WACCM SAI-REF) and the model-mean SAI response (CAM SAI-CNT + WACCM SAI-CNT)/2. Low values (much smaller than 1) indicate desired results. All data are averaged over [2016–2035] (reference) or [2080–2099] (control and SAI2020). Hatching indicates where the intermodel mean SAI response is smaller than 1 K (left) and 2.5 K (right), i.e. where the index is less reliable. These values are somewhat arbitrary, choosing larger (smaller) values will make the hatched area larger (smaller).

**Figure A5.** CAM-WACCM model differences of potential temperature (top) and zonal mean zonal wind (bottom). All data are averaged over [2016–2035] (reference) or [2080–2099] (control and SAI2020). Black contours in the temperature maps indicate the corresponding values in WACCM for each experiment.

## Appendix B: Aerosol distribution

**Figure B1.** Annual mean global mean aerosol optical depth (a) and prescribed zonal mean sulphate aerosol mass concentration (b) in CAM-SAI2020 (shading) and WACCM-SAI2020 (black contours). Aerosol optical depth represents the stratospheric component of aerosol optical depth at 550 nm (day,night), except for WACCM-control where it is the AOD at 550 nm due to sulphate aerosol (for data availability reasons). The zonal mean sulphate concentration is averaged over [2080–2099] in both CAM and WACCM, and has units  $10^{-7}$  kg/kg.

# B1 Examining the relevance of dynamically scaling ozone

400

405

Ozone concentrations in WACCM increase significantly with time, most likely due to a reduction in ozone-depleting substances (Fig. B2a). Maximum zonal mean ozone concentration is higher in WACCM-SAI2020 than WACCM-Control, though the difference is small relative to the increase from 2020 to 2100 in both experiments. In CAM, ozone is prescribed following the SSP5-8.5 scenario, which is practically identical to ozone in WACCM-Control.

A more detailed view of the zonal mean ozone difference between WACCM-SAI2020 and WACCM-Control is shown in Fig. B2b. In the lower stratosphere, ozone change follows a slightly complicated pattern. As a response to SAI In WACCM simulations with interactive ozone chemistry, Richter et al. (2017) find stratospheric ozone concentration increases above the sulphate aerosol layer, while it decreases inside, with a reversed pattern on the opposite hemisphere for hemispheric injection. This pattern may be recognized in the current reference WACCM simulation given more aerosols are injected on the Southern Hemisphere than the Northern Hemisphere (Fig. B1b), though this is just an observation and further explanation is beyond the scope of this study. In the upper stratosphere, ozone concentrations generally decrease due to SAI in WACCM, most notably at the poles. These effects naturally lack in CAM, because ozone is prescribed by the SSP5-8.5 scenario. As a consequence, upper-stratospheric polar ozone concentrations in CAM-SAI2020 are higher than in WACCM-SAI2020 (Fig. B2c,f).

The increase of stratospheric ozone concentration with time is much larger than its response to SAI (Fig. B2b,d,e). Note that, because prescribed stratospheric ozone in CAM is virtually identical to stratospheric ozone in WACCM-Control, (b) may be

interpreted as the difference (e) minus (d) and (d) as the change in ozone from the reference to the control period in CAM- and WACCM-Control. This increase contributes to warming above the aerosol layer by increased radiation absorption, however the effect of radiative cooling due to increases greenhouse gas concentrations is stronger (Fig. 7). Hence, the impact of SAI on ozone is relatively minor compared to the effect of reducing ozone-depleting substances in SSP5-8.5, and the effect on climate is likely small. Furthermore, the effect of model differences on stratospheric temperature and circulation is much stronger than the effect of ozone, resulting in significantly colder polar stratospheric temperatures and stronger stratospheric jets in CAM (Fig. A5).

Figure B2. Changes in zonal mean and polar mean ozone. (a) evolution of maximum zonal mean ozone, (b) zonal mean ozone response to SAI in WACCM, (c) model difference of interannual mean ozone averaged over [60N–90N], (d) temporal change of zonal mean ozone in CAM-SAI2020, (e) temporal change of zonal mean ozone in WACCM-SAI2020, (f) model difference of interannual mean ozone averaged over [60S–90S]. Contours indicate mean ozone for CAM-Control (b), CAM-Reference (d,e) and WACCM-SAI2020 (c,f). WACCM-Control data is available only on 19 pressure levels with relatively poor resolution in the upper stratosphere, whereas all other data is provided on 70 vertical levels. Ozone in CAM is prescribed by the SSP5-8.5 scenario, which is virtually identical to WACCM-Control. Therefore, CAM data is substituted for WACCM-Control data (indicated with a superscript C) in all panels except (a), where the coarse levels are used. In (a), the maximum of the zonal mean is used because the coarse levels do not extend into the upper stratosphere (<3 hPa), making the maximum a more robust representation of the field than a vertically integrated value.

## 420 Appendix C: Simulation with CESM1

While the main results are shown only for CESM2(CAM)6 simulations, similar experiments have been performed CESM1.0.4(CAM5). For this configuration, the regressions are performed using data from the Geoengineering Large ENSemble (GLENS) simulations with CESM1(WACCM) (Tilmes et al., 2018). An overview of these simulations is presented in Table 1 and methods and results from the regression analysis are shown in Table C1.

Table C1. Interannual normalization definitions CESM1

| $\overline{F_i}$ | $n_i(y)$                                 | $c_0$           | $c_1$          | $c_2$      |
|------------------|------------------------------------------|-----------------|----------------|------------|
| AODVISstdn       | $\int \text{AODVISstdn}  \mathrm{d}S$    | -               | -              | -          |
| so4_a1           | $\int \text{so4\_a1}\mathrm{d}V$         | 1.717179696e9   | 3.814912693e9  | 0.65606465 |
| so4_a2           | $\int \text{so4}_{-}\text{a2}\text{d}V$  | -3.042262144e-8 | 4.863942358e-8 | 0.16051887 |
| so4_a3           | $\int \text{so4}_{-} \text{a3}  dV$      | -1.373741787e9  | 1.708376730e11 | 1.14287290 |
| H2SO4_mass       | $\int {\rm H2SO4\_mass}{\rm d}V$         | -3.164908547e26 | 1.387540834e29 | 1.16797461 |
| H2SO4_mmr        | $\int \mathrm{H2SO4}$ _mmr d $V$         | 7.776151719e8   | 1.978583958e11 | 1.13329202 |
| REFF_AERO        | $\int \mathrm{REFF\_AERO}^2 \mathrm{d}V$ | -6.660152110e12 | 1.259288046e14 | 0.60356471 |
| rmode            | $\int \mathrm{rmode}^2  \mathrm{d}V$ ]   | -6.660152203e10 | 1.259288047e12 | 0.60356471 |
| SAD_AERO         | $\int SAD_AERO^2 dV$                     | 6.900165392e-6  | 1.420431732e-4 | 0.90575559 |
| sad              | $\int \operatorname{sad}^2 dV$           | 6.900170210e2   | 1.420431708e4  | 0.90575567 |

 $<sup>\</sup>int \dots \mathrm{d}S$  = annual mean global surface mean,  $\int \dots \mathrm{d}V$  = annual mean mass-weighted volume integral over the entire atmosphere

Unlike the CESM2(CAM)6 configuration, the CESM1 configurations require modifications to the applied forcing as the prescribed volcanic aerosol has one size mode, whereas the CESM1(WACCM) output has three modes. To convert from three modes to one, regressions are performed on each of the CESM1(WACCM) modal variables, though only mass mixing ratio is required for CESM1(CAM) (wet diameter is 'fixed'). Then, every model year the modal mass mixing ratios are calculated using eq. 1 in section 2.2 and summed to get the mass mixing ratio of volcanic aerosol.

# 430 C1 CESM1 Climate Results

Figure C1. Annual mean anomalies of the global mean temperature,  $T_0$ , (a), the interhemispheric gradient of surface temperature,  $T_1$ , (b), and the equator-to-pole gradient of surface temperature,  $T_2$ , (c) for CESM1.0.4. The anomalies are defined with respect to the [1990–2009] mean CAM5-Reference (perpetual 2000 conditions spinup, light blue) data, giving  $T_{0,\rm ref}=288.92~{\rm K}, T_{1,\rm ref}=0.53$  and  $T_{1,\rm ref}=-11.32$ .

**Figure C2.** Similar to Fig. C1, but for high-res CESM1. The reference values are  $T_{0,\text{ref}} = 288.08 \text{ K}$ ,  $T_{1,\text{ref}} = 0.85$  and  $T_{1,\text{ref}} = -11.47$ .

**Figure C3.** [2075–2095] surface temperature anomalies (top) and precipitation anomalies (bottom) for CESM1. The anomalies are shown for CAM (left) and HR-CAM (right), having a nominal resolution of 1 and 0.5 degrees, respectively. Contours represent the reference values, which are drawn at 0.5 (red, hatched) and 4 (blue) mm/day for precipitation. The reference period is [1990–2009].

Figure C4. Similar to Fig. C3, but showing the zonal means.

Author contributions. JJ worked on the original draft and editing, validation, formal analysis, methodology, visualization. DP worked on the methodology, conceptualization, original draft, formal analysis, validation, visualization. SL worked on the formal analysis, visualization and draft review. CW worked on conceptualization, supervision, methodology, original draft, draft review, project administration, funding acquisition. MB worked on conceptualization, supervision, draft review, methodology, project administration and funding acquisition. RW worked on the software, data curation and draft review

Competing interests. The authors declare there are no competing interests.

435

Acknowledgements. This research is funded by the Dutch Ministry for Education, Culture and Science through the Van Meenen funds (16604027) and via the Sectorplan Science and Technology. We thank Leo van Kampenhout for his help in setting up and running simulations. We thank Michael Kliphuis for his help in addressing technical issues.

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
