# Peer review of "A computationally efficient method to model Stratospheric Aerosol Injection experiments"

_EGUsphere, 2025_

## Author Comment (AC1)

RC1 - This is a fantastic paper. The work that the authors have done has the power to make complex geoengineering simulations more accessible. The development and exploration of the methodology are done quite well. I do have some comments:
AC - We are pleased to hear you think our work matters and is performed quite well.

RC1 - I don't have a sense of pros and cons, i.e., when this simpler method would work versus when you need a more complex model like WACCM. I don't expect anything thorough, but if the authors could provide some opinions on this, it would be helpful.
AC - Yes, that is a good suggestion. We will add our opinion on this in the revised version. Roughly speaking, users more interested in tropospheric or ocean processes who need to limit computational costs may want to use our method, whereas researchers interested in stratospheric circulation or microphysics ought to use WACCM.

RC1 - There are situations where the authors claim to have explored a variety of scenarios and conclude that the method is robust to different scenarios. This is only partially true. You may get different answers if you use a different background scenario, for example one with strong mitigation or changes in tropospheric aerosols, as that will change the spatial patterns of forcing. Some appropriate caveats would be useful.
AC - Indeed, we have only considered changes in the aerosol forcing scenarios, not the background greenhouse gas forcing scenarios. We will ensure that this is evident from our statements and add caveats where appropriate.

RC1 - Lines 152-153: I think providing more details about the simulations here would be useful. I got a little lost. I suggest moving Table C1 into the main body of the text and expanding it so that it has more information about the specifications of each simulation.
AC - For better overview, we will summarise all simulations in an expanded table in the main.

RC1 - Bullets on page 3: This is essentially pattern scaling. There's a lot of literature you can lean on to show that this is a sensible thing to do.
AC - Agreed. We will add a couple of references here to back up our motivation for using this technique.

RC1 - Line 101: What does "similar" mean?
AC - This is quite a general statement indeed. What we roughly meant is a similar response in surface temperature, precipitation, tropospheric winds and temperature to SAI. We will add this to the manuscript.

RC1 - Line 186: I think you mean monotonically?
AC - Yes, we will fix this.

RC1 - Section 2.2: It would be useful if you said somewhere that the approximations you make are good enough for this purpose, as the point of a feedback algorithm is to correct for such uncertainties. MacMartin et al. and Kravitz et al. both say this if you need citations.
AC - Indeed, the original feedforward/feedback algorithm is constructed in such a way as to be very robust w.r.t. suboptimal parameter choices. We found that even for rather suboptimal choices (e.g., a much too strong feedforward due to model differences in sensitivity), errors in the temperatrue target were limited (order 0.3 degrees), though noticeable. We will stress this robustness in the revised manuscript.

RC1 - Line 241: Your errors seem kind of high. Looking at Figure 2, an error of 0.4°C is a lot. This likely means your controller isn't tuned as well as it could be, which isn't a big deal, but it would be worth saying so.
AC - The relatively high error is due to the fact that the algorithm in its original form (feedback with

integrator and proportional term) is not constructed to deal with sudden changes, such as starting strong SAI in 2080. Rough "dry-testing" experiments with a simple box model emulator for GMST suggest that this can not be resolved by simple re-tuning. Therefore we have made an ad-hoc improvement by resetting the integrated error term, as described in the manuscript, which reduces but does not eliminate the "cooling overshoot" observed in fig. 2b. We will state this more explicitly in the revised manuscript.

RC1 - Figure 2: I found the panels confusing, in that you're mixing and matching units.
AC - We understand the confusion. To improve the figure we will add separate y-axes in blue for the temperature errors. This will avoid the mixing of units.

RC1 - Line 257: Per the above comment, maybe change "well" to "adequately".
AC - Yes, we will change this.

RC1 - Line 292: This is correct but also a strawman argument. You didn't try to restore the climate completely.
AC - Thank you for pointing this out. We will rephrase this more factually such that we still acknowledge local differences without pretending it to be our goal to restore the climate completely.

RC1 - Figure 5: I'm having trouble making sense of how important these results are. I wonder if you could compute z-scores (or something like that) so I would know whether the CAM minus WACCM differences are large compared to the natural variability of WACCM.
AC - Good point. We will add hatching to the revised figure 5 in the manuscript to indicate significance. Additionally, hatching will be applied to other figures that we think will benefit from this.

RC1 - I did not find the paragraph on lines 376, nor Appendix B, terribly convincing. If you heat the stratosphere by 24°C, you are going to have substantial influences on ozone, and we know that ozone has an influence on surface climate. I would be more comfortable if you simply said that this is what you did, its effects of ozone changes on climate are likely smaller than the effects of the stratospheric heating on climate, and this should be explored further. That puts you on much safer ground.
AC - We initially included this section to provide some more background into the matter, but this indeed poses a risk as the processes involved are more complex than shown here, requiring further analysis. As our article is meant to convey a useful protocol for practitioners, it would indeed be safer to state the things you mention. We will do so in a revised version of the mannuscript.

RC1 - Lines 400-402: Kravitz et al. (2014) demonstrated that the controller is likely robust to these sorts of differences. That gives some confidence that your controller indeed can handle this.
AC - That is a good suggestion. We will mention this in the revised text.

RC1 - Lines 427ff: See recent work from the Cornell group, specifically led by Farley or Brody. They're doing the initial steps of what you propose.
AC - We will read their relevant publications and refer to these in the outlook.

RC1 - Code availability: Journals tend to want a fixed repository (e.g., Zenodo) rather than a changeable repository (Github).
AC - We will make a Zenodo repository for the revised version.

---

## Author Comment (AC2)

*This manuscript by de Jong at al. is trying to do two things: first, it is trying to describe how to prescribe stratospheric aerosols forcing in a different model. Second, it's trying to show the surface climate differences in two different CESM1 versions (CESM2-CAM6, CESM1 and CESM2-WACCM6) forced by the same aerosol fields (prescribed, in the case of CAM).*

*The main thing I take issue with, in the first part, is that essentially what they are proposing to do is not novel at all. Prescribing an aerosol field externally is something that has been done before multiple times not just for volcanism (the entirety of CMIP6 historical simulations used prescribed aerosols) but for SRM, too. Conveniently, the authors leave out all the references of this, trying to pass it off as "novel". Personally, I deeply dislike this way of not acknowledging past work (or to avoid performing a literature search that would have made past works emerge) to hype up one's own.*

*Here are three main examples:*

- *Tilmes et al. (2016) proposed a prescribed aerosol forcing file for models with no interactive sulfate cycle. Such forcing field was then used in Xia et al. (2017) and in one of the models for G6sulfur (CNRM) as described in Visioni et al. (2021).*

- *Still in G6sulfur, the model MIP prescribed their own aerosol field and scaled them up in their two fully-coupled versions, also as described in Visioni et al. (2021).*

- *Finally, Tilmes et al. (2024) proposed a new experiment for CCMI with a new aerosol field for SAI, also describing climatic differences in WACCM when fully interactive vs prescribed aerosols are used, with much more details provided on how to prescribe the aerosols fields in different models provided in Jörimann et al. (2025). Reading Jörimann et al. (2025) could also illuminate for the authors how hard it is to prescribe aerosol fields in other models' version, as one needs to get optical properties that might be treated differently in different models with different ways to translate aerosol size distribution to forcing.*

*None of this is ever acknowledged in the (rather short and non comprehensive in general) cited literature, giving the reader the impression that this is the first time something like this has been attempted (cue the word "novel" used frequently in the text, and also in the Key Points).*

*So, what's novel? The use of a scaled-up, mono-dimensional control algorithm (not that different from what was used in multiple G6 models, see above) used in multiple publications starting in MacMartin et al. (2014)? That's hardly new – and the simplified controller used here hardly seem well tuned, considering the pretty large errors shown in Fig. 2!*

We thank the reviewer for the extensive comments and literature suggestions. However, we suspect there was a misunderstanding about the scope of our work, and in particular, what aspects thereof we consider to be "novel". At no point has it been our intention to pretend that we came up with the idea to force models with aerosol fields, either directly or scaled similarly as in previous work, and it never occurred to us that our text might be read in this way. Reviewer 1 seems to confirm that this is a reasonable assumption.

Instead, our paper intends to present and test a useful protocol for a specific application of forcing climate models with aerosol fields, namely, mimicking GLENS / ARISE-like SAI simulations in situations where running a full stratospheric chemistry model is impractical, for example due to computational constraints. For this specific application, our method is novel to our best knowledge, and modellers from several groups expressed interest in using it.

The submitted manuscript aims to disseminate our protocol and outline potential avenues for future use, which fits within the stated purposes of GMD ("development and technical papers, describing

developments such as new parameterizations or *technical aspects of running models*" and "model experiment descriptions, including experimental details and project protocols".

We will take care to clarify our intentions. We will pay particular attention to reformulating the abstract, which we agree can be read as if the scope of our work is broader than it is. Please find further specific changes we intend to apply to the manuscript below.

**Relation of our protocol with previous work**

From preliminary tests we found that simply inserting WACCM-derived aerosol fields in CAM will lead to rather different temperature outcomes (i.e., an aerosol field taken from WACCM cannot stabilise GMST in CAM). Such a forcing can be scaled manually, but this can be cumbersome, especially when running non-standardized SAI forcing scenarios. Hence, for our application of mimicking WACCM forcing in CAM we explored the possibility of using a feedback controller combined with a new method of scaling. Our protocol builds on two prior strands of work:

- SRM simulations with feedback controllers. For us, SAI simulations with a full-fledged chemistry model are most relevant, although feedback controllers have also been applied on simpler simulations and are currently being applied to Marine Cloud Brightening. Examples involving SAI include the simulations with CESM(WACCM) for G6sulfur (Visioni et al. 2021), GLENS (Kravitz et al., 2017), and an inclusion of the same feedback controller in the ARISE-SAI protocol (Richter et al., 2022).

- Prescribing (stratospheric) aerosol forcing to models that do not simulate them explicitly; for us, applications in SAI are the most relevant, but others exist (volcanic eruptions). Examples include the CNRM and MPI-ESM models that participated in G6sulfur (Visioni et al., 2021), which may be forced by the aerosol forcing set provided by Tilmes et al., 2015 (CNRM) or a similar set produced by the same model (MPI-ESM) and scaled by means of choice, CESM(CAM-chem) (Xia et al., 2017), and an inclusion of using prescribed forcing data in an SAI protocol for CCMI (Tilmes et al., 2024). Regarding this protocol, the REMAPv1 code (Jörimann et al., 2024) is proposed as a viable method to calculate aerosol properties if the input dataset does not contain the fields required by the model.

Our method adds to the existing literature in two ways:

- Most importantly: Combining a feedback controller with aerosol forcing

- Allowing for non-linear relationships between AOD and other aerosol variables

**Combining a feedback controller with aerosol forcing.** In the literature mentioned above and suggested by the reviewer, we have not encountered a publication documenting the combined use of a feedback controller and prescribing aerosols.

Tilmes et al. (2015) provides a predetermined forcing scenario, which is directly prescribed to the model in Xia et al. (2017). Both CNRM and the two versions of MPI-ESM in Visioni et al. (2021) scale such prescribed forcing by multiplying the AOD with some year-dependent factor to achieve the temperature target. Automatising the feedback controller - as has been done in e.g. GLENS and ARISE with full aerosol models - obviously can save considerable time, especially when exploring new scenarios in the simpler model setup (as out SAI2080 in CESM-CAM).

In short, one original contribution of our paper is simply to have worked out a practical way of combining these previous strands of work in order to allow SAI simulations in the (computationally cheap) CAM to mimic SAI simulations in WACCM as closely as possible in terms of (tropospheric / near-surface) climate impacts, here using GMST as target variable - and to document it sufficiently clearly

(including some practical details such as the iteration to get a sufficiently good feedforward) that other groups can hopefully save themselves time and errors if they wish to implement something similar.

**Nonlinear relationship between AOD and other variables.** Apart from using a feedback controller for scaling aerosol forcing - more precisely, AOD - we also use non-linear fits to relate other variables such as aerosol mass, size and surface area to AOD. As far as we can see from the literature, this was not the case in previous studies where scaled aerosol forcing was used. A consequence of this approach is that aerosol field properties may not change when scaling the forcing strength. Tilmes et al. (2022) for example find that a linearly increasing forcing strength for CNRM in G6sulfur resulted in a linearly increasing SAD because of the fixed aerosol size, which is most likely dependent on the forcing strength (Niemeier et al. 2011, Niemeier and Timmreck 2015). Though the relationship between AOD and the dominant mode aerosol mass is fairly linear in our simulations, other variables show different relations (fig. 1 of the manuscript).

**Translating aerosol fields for use in different models** The prescription of aerosols into models that require different aerosol properties than provided requires the use of Mie calculations as done by the REMAPv1 algorithm developed by Jörimann et al. (2025). In our analyses, we did not need such translations as WACCM output may be directly prescribed to CAM. Our method can therefore be applied by anyone using a compatible pair of models, regarding the stratospheric forcing, but not when different input is required. In our revised version of the manuscript, we will stress that we have not validated the case for non-matching input fields, e.g. with a less general description in the abstract. It might be possible to combine the REMAPv1 algorithm with our method if one needs to translate between incompatible pairs of models (i.e., the model receiving forced aerosols requires different inputs than the model generating the forcing). We will mention this in the discussion.
Summarizing, the proposed method uses an improved scaling not seen before in prescribed-aerosol SAI modelling and can easily be combined with existing feedback controllers. This makes it a valuable tool and source of inspiration for modelling groups that will perform their own SAI experiments (focussed on impact).

We will add the above discussion on prior works and how our work relates to them to the revised manuscript.

*The second part then offers a description of surface climate differences between CAM and WACCM. This part (which would also not be particularly fitting for GMD on its own) is awfully lacking as well.*

*First, I don't see any kind of discussion of the biases between the two models' versions without SAI, and I think that would be a fundamental starting point to understand the sources of differences to a forcing. Even very similar versions of CESM2-WACCM6 that differ by the inclusion or not of tropospheric chemistry can present relevant distinctions in some modeled trends (see i.e. Davis et al., 2023), let alone two basically different models. Aside from surface climate, most of the differences in the stratospheric response are also left very vague (for instance, Section 3.3 essentially doesn't offer anything more than guesses ("These features occur in Control as well and are assumed to be model differences") that don't really provide much confidence to the reader that the authors know what they're talking about. The entire paragraph from l. 376 to l. 384 is very high level, does not cite any of the relevant literature about the impacts of stratospheric ozone changes on surface climate (to begin with, but the list is very long, see Bednarz et al., 2022).*

*I do not wish to claim that novelty in itself makes a study worthy of being published, and actually I think quite the contrary: but a study that wishes to claim novelty as its main strength, fails to deliver and also ignores relevant literature that would help frame this work in the broader context (out of cu-*

*riosity, the manuscript has 24 references, of which 7 are in the first paragraph to prove/disprove the permissibility of SRM research or to discuss climate change, one of them is a non-peer reviewed link to CarbonBrief, and one a never accepted preprint from 2023, casually including one of the authors of this piece) is setting itself up to fail for anyone who has any familiarity with the field.*

**Literature review - additional points**

We thank the referee for pointing out the Bednarz et al., 2022 paper, which can serve as a useful comparison case.
We would like to keep the initial, more general remarks about the controversies around SRM, because we believe that even an otherwise technical paper should provide a minimum of context regarding SRM's controversies and dilemmas. The Futerman et al., 2023 paper is accepted and we expect to have the link to the final version by the time this manuscript is published. The Pflüger et al., 2024 reference is very relevant because the same forcing method is used and is correctly referring to a published peer-reviewed paper.

**Validation**

We now address concerns raised by referee 2 regarding the quality of our validation. What we present is a tool that essentially emulates the stratospheric aerosol forcing in a model that does not have stratospheric chemistry. The purpose of its validation is to show how well the tropospheric climate responds to this applied forcing, so that modellers may decide to use our proposed method or not.
For our validation we used the following metrics (in decreasing order of importance:

- Can the method achieve the set climate target (GMST)?
  GMST can be stabilised to the desired level, which is the main objective of our protocol. This works nicely when reproducing WACCM scenarios (in our case, SAI2020). When generating new scenarios which strongly differ from the WACCM scenarios used as input (in our case, SAI2080), discrepancies occur but these are limited and explainable and can be reduced further by fine-tuning the procedure (in our case, resetting the integrator term of the feedback) .

- Does the method reproduce the response to SAI of the tropospheric climate?
  In particular we verify if (surface) temperature, precipitation and wind responses to SAI in CAM resemble responses in WACCM for equivalent scenarios. The model discrepancies we find are not larger than inter-model difference in the absence of SAI. Inter-model differences are evaluated in Reference and RCP8.5. Please note that the scope of our validation is merely to check whether our method introduces large additional biases (relative to existing inter-model differences in the absence of SAI), not to provide an in-depth comparison between CAM and WACCM more broadly.

- Does the CAM simulation successfully control all degrees of freedom targetted in WACCM, even if not included in CAM's feedback controller? Does this depend on changes in the scenarios (e.g. SAI2080 instead of SAI2020) or model version (e.g. switching to higher resolution)?
  We find that even if we only control for GMST, other targets used in the unerlying wACCM simulations (T1, T2) are fairly well reproduced if CAM is used to generate a similar scenario (SAI2020). The same does not hold for the rather extreme SAI2080 scenario. This discrepancy can be explained by the long-term AMOC changes incurred during the prolonged warming under SAI2080. As we also discuss in the manuscript, this discrepancy could likely be mitigated by expanding our protocol with additional degrees of freedom, i.e. including several aerosol patterns associated with different injection latitudes in WACCM. We agree with the reviewer that it would be interesting to test this. However, for the time being, we are restricted to using for the validation the simulations we have now (which were partly developed for other pieces of analysis), therefore

we cannot do more than discussing this (current) limitation and suggesting an expansion to multiple aerosol patterns as an avenue for future development.

- Stratosphere:
  Our method should not be used by anyone looking for high fidelity in the stratosphere, as mentioned in our abstract and introduction. For such research interest, clearly using WACCM is a more viable approach. Our interpretations on the differences in the stratosphere such as in section 3.3 are considered to be a nice-to-have and do not comprise an in-depth analysis of the mechanisms resulting in the observed patterns of change. They are a simple check whether the limited representation of stratospheric processes - such as atmosphere-ozone interaction or other things that could affect stratospheric heating - should be expected to have impacts in tropospheric circulation (as for example suspected by McCusker 2015)

Regarding the above, we believe that the validation in the current version of the manuscript is fit for purpose. We agree that we should avoid potential misunderstandings about the range of problems out methods should, or shouldn't, be applied to. In the revised version of the manuscript, we will state more clearly that our method is not to be used for prescribing aerosols in SAI simulations where high fidelity in the stratosphere is required, but rather as a tool for simulations meant to analyse the tropospheric/oceanic impacts of geoengineering.

*This manuscript, in another venue but GMD, could provide some interesting analyses of what happens when you prescribe the same aerosol field in different model's versions given much more in depth, careful analyses, exploring the sensitivities, performing further experiments (i.e. different scenarios, which are available in CESM2-WACCM6, could be compared with the same prescribed CAM6 results to understand if the differences observed are a function of magnitude, of the specifics of the AOD pattern, of the broader atmospheric response, of the underlying emission scenario; the ozone field could have actually been changed; CAM6 simulations with fixed SSTs could have been performed to understand the actual forcing response; different patterns of AOD from different injection locations could have been combined to understand the linearity of the response, etc. The authors do acknowledge that "It might be worthwhile to expand the method for additional use cases." But, given the paucity of interesting results otherwise, I think they should be the ones, and not somebody else at a later date). In its current form, however, I cannot recommend publication in this venue.*

**On the additional suggested experiments**

A long list of additional simulations and analyses is suggested by referee 2 to develop a deeper understanding of the stratospheric forcing response. We have no doubt that many of these would greatly contribute to the community's understanding of stratospheric aerosol processes and aerosol-induced forcing. However, the scope of this paper is not a detailed research aerosol processes related to SAI or provide an in-depth model comparison, but to present a useful protocol for practitioners interested in (surface) climate impacts. Therefore the suggested fixed-SST experiment, while very interesting for wider research on aerosol forcing, are beyond the scope of this study.

We agree that some other suggested experiments would be nice to add here. For example, as regards additional scenarios, it would be nice, for the purpose of additional validation, to try using CESM-CAM aerosol fields based on a particular CESM-WACCM simulation (e.g. what we call SAI2020 here) to reproduce another existing CESM-WACCM simulation (e.g., ARISE-like scenarios (Richter et al., 2022)). Also, as discussed above, it would be nice to expand our method by using aerosol patterns from several injection latitudes to test for linearity and improve the performance of new scenarios (here SAI2080) on additional degrees of freedom. Unfortunately, we have to make do with simulations originally designed for other research projects. Adding the suggested new simulations would require a very substantial amount of additional computational and human-time effort, which we cannot commit

and which do not weigh up against the gains achievable for this paper. We therefore do not intend to perform the suggested additional experiments.

**References**

Bednarz, E. M., Visioni, D., Banerjee, A., Braesicke, P., Kravitz, B., & MacMartin, D. G.: The overlooked role of the stratosphere under a solar constant reduction. Geophysical Research Letters, 49, e2022GL098773. `https://doi.org/10.1029/2022GL098773`, 2022.

Davis, N. A., Visioni, D., Garcia, R. R., Kinnison, D. E., Marsh, D. R., Mills, M., et al.: Climate, variability, and climate sensitivity of "Middle Atmosphere" chemistry configurations of the Community Earth System Model Version 2, Whole Atmosphere Community Climate Model Version 6 (CESM2(WACCM6)). Journal of Advances in Modeling Earth Systems, 15, e2022MS003579. `https://doi.org/10.1029/2022MS003579`, 2023.

Jörimann, A., Sukhodolov, T., Luo, B., Chiodo, G., Mann, G., and Peter, T.: A REtrieval Method for optical and physical Aerosol Properties in the stratosphere (REMAPv1), EGUsphere [preprint], `https://doi.org/10.5194/egusphere-2025-145`, 2025.

Kravitz, B., Robock, A., Tilmes, S., Boucher, O., English, J. M., Irvine, P. J., Jones, A., Lawrence, M. G., MacCracken, M., Muri, H., Moore, J. C., Niemeier, U., Phipps, S. J., Sillmann, J., Storelvmo, T., Wang, H., and Watanabe, S.: The Geoengineering Model Intercomparison Project Phase 6 (GeoMIP6): simulation design and preliminary results, Geosci. Model Dev., 8, 3379–3392, `https://doi.org/10.5194/gmd-8-3379-2015`, 2015.

Kravitz, B., MacMartin, D. G., Mills, M. J., Richter, J. H., Tilmes, S., Lamarque, J.-F.,... Vitt, F.: First simulations of designing stratospheric sulfate aerosol geoengineering to meet multiple simultaneous climate objectives. Journal of Geophysical Research: Atmospheres, 122, 12,616–12,634. `https://doi.org/10.1002/2017JD026874`, 2017.

McCusker, K. E., Battisti D. S., and Bitz C. M.: Inability of stratospheric sulfate aerosol injections to preserve the West Antarctic Ice Sheet. Geophys. Res. Lett., 42, 4989–4997 `https://doi.org/10.1002/2015GL064314`, 2015.

Niemeier, U., Schmidt, H. and Timmreck, C.: The dependency of geoengineered sulfate aerosol on the emission strategy. Atmosph. Sci. Lett., 12: 189-194. `https://doi.org/10.1002/asl.304`, 2011.

Niemeier, U. and Timmreck, C.: What is the limit of climate engineering by stratospheric injection of SO2?, Atmos. Chem. Phys., 15, 9129–9141, `https://doi.org/10.5194/acp-15-9129-2015`, 2015.

Richter, J. H., Visioni, D., MacMartin, D. G., Bailey, D. A., Rosenbloom, N., Dobbins, B., Lee, W. R., Tye, M., and Lamarque, J.-F.: Assessing Responses and Impacts of Solar climate intervention on the Earth system with stratospheric aerosol injection (ARISE-SAI): protocol and initial results from the first simulations, Geosci. Model Dev., 15, 8221–8243, `https://doi.org/10.5194/gmd-15-8221-2022`, 2022.

Tilmes, S., Mills, M. J., Niemeier, U., Schmidt, H., Robock, A., Kravitz, B., Lamarque, J.-F., Pitari, G., and English, J. M.: A new Geoengineering Model Intercomparison Project (GeoMIP) experiment designed for climate and chemistry models, Geosci. Model Dev., 8, 43–49, `https://doi.org/10.5194/gmd-8-43-2015`, 2015.

Tilmes, S., Visioni, D., Jones, A., Haywood, J., Séférian, R., Nabat, P., Boucher, O., Bednarz, E. M., and Niemeier, U.: Stratospheric ozone response to sulfate aerosol and solar dimming climate interventions based on the G6 Geoengineering Model Intercomparison Project (GeoMIP) simulations, Atmos. Chem. Phys., 22, 4557–4579, https://doi.org/10.5194/acp-22-4557-2022, 2022.

Tilmes, S., Bednarz, E. M., Jörimann, A., Visioni, D., Kinnison, D. E., Chiodo, G., and Plummer, D.: Stratospheric Aerosol Intervention Experiment for the Chemistry-Climate Model Intercomparison Project, EGUsphere [preprint], `https://doi.org/10.5194/egusphere-2024-3586`, 2024.

Visioni, D., MacMartin, D. G., Kravitz, B., Boucher, O., Jones, A., Lurton, T., Martine, M., Mills, M. J., Nabat, P., Niemeier, U., Séférian, R., and Tilmes, S.: Identifying the sources of uncertainty in climate model simulations of solar radiation modification with the G6sulfur and G6solar Geoengineering Model Intercomparison Project (GeoMIP) simulations, Atmos. Chem. Phys., 21, 10039–10063, `https://doi.org/10.5194/acp-21-10039-2021`, 2021.

Xia, L., Nowack, P. J., Tilmes, S., and Robock, A.: Impacts of stratospheric sulfate geoengineering on tropospheric ozone, Atmos. Chem. Phys., 17, 11913–11928, `https://doi.org/10.5194/acp-17-11913-2017`, 2017.

---

## Author Comment (AC3)

Dear editor,

We thank you for checking the compliability of our manuscript and acknowledge you have a fair point regarding the reusability of our code and data.

To make our work comply with your code and data policy, we will upload the used CESM(CAM) code, feedback controller code and WACCM analysis code as well as links to the relevant data to the repository and publish the repository in a permanent store with a DOI.

We hope this is sufficient to solve the issue.

Kind regards,

Jasper de Jong (on behalf of all authors)

---

## Author Comment (AC7)

RC1 - This is a fantastic paper. The work that the authors have done has the power to make complex geoengineering simulations more accessible. The development and exploration of the methodology are done quite well. I do have some comments:
AC - We are pleased to hear the reviewer thinks our work matters and is performed quite well!

RC1 - I don't have a sense of pros and cons, i.e., when this simpler method would work versus when you need a more complex model like WACCM. I don't expect anything thorough, but if the authors could provide some opinions on this, it would be helpful.
AC - Yes, that is a good suggestion. We will add our opinion on this in the revised version. Roughly speaking, users more interested in tropospheric or ocean processes who need to limit computational costs may want to use our method, whereas researchers interested in stratospheric circulation or microphysics ought to use WACCM.
AC - We have limited the use of our method to climate impact analyses in the troposphere below. A few remarks are mentioned in L9-11, L36-37, and more details on use cases are provided in L350ff.

RC1 - There are situations where the authors claim to have explored a variety of scenarios and conclude that the method is robust to different scenarios. This is only partially true. You may get different answers if you use a different background scenario, for example one with strong mitigation or changes in tropospheric aerosols, as that will change the spatial patterns of forcing. Some appropriate caveats would be useful.
AC - Indeed, we have only considered changes in the aerosol forcing scenarios, not the background greenhouse gas forcing scenarios. We will ensure that this is evident from our statements and add caveats where appropriate.
AC - We added "SAI" to scenarios and mentioned the caveats in: L13, L319, L373-377.

RC1 - Lines 152-153: I think providing more details about the simulations here would be useful. I got a little lost. I suggest moving Table C1 into the main body of the text and expanding it so that it has more information about the specifications of each simulation.
AC - For better overview, we will summarise all simulations in an expanded table in the main.
AC - We have placed all the simulations in Table 1. in the main body, additionally we created a schematic diagram to illustrate the different experiments (Fig. 1).

RC1 - Bullets on page 3: This is essentially pattern scaling. There's a lot of literature you can lean on to show that this is a sensible thing to do.
AC - Agreed. We will add a couple of references here to back up our motivation for using this technique.
AC - We have added some background on pattern scaling in L46-52. Moreover, we included lots of other research in the introduction to establish a better view of how our works relates to them.

RC1 - Line 101: What does "similar" mean?
AC - This is quite a general statement indeed. What we roughly meant is a similar response in surface temperature, precipitation, tropospheric winds and temperature to SAI. We will add this to the manuscript.
AC - The reviewer probably meant we should make a more concrete statement concerning when we think results are similar. We moved this part to the start of the results section and provided a new statement in L198-200. (In the preceeding lines, we mention what variables we will look at as well.)

RC1 - Line 186: I think you mean monotonically?
AC - Yes, we will fix this.
AC - We have changed this in L138-139.

RC1 - Section 2.2: It would be useful if you said somewhere that the approximations you make are

good enough for this purpose, as the point of a feedback algorithm is to correct for such uncertainties. MacMartin et al. and Kravitz et al. both say this if you need citations.

AC - Indeed, the original feedforward/feedback algorithm is constructed in such a way as to be very robust w.r.t. suboptimal parameter choices. We found that even for rather suboptimal choices (e.g., a much too strong feedforward due to model differences in sensitivity), errors in the temperatrue target were limited (order 0.3 degrees), though noticeable. We will stress this robustness in the revised manuscript.

AC - We have mentioned that the feedback controller is robust to these discrepancies in L155-157.

RC1 - Line 241: Your errors seem kind of high. Looking at Figure 2, an error of 0.4°C is a lot. This likely means your controller isn't tuned as well as it could be, which isn't a big deal, but it would be worth saying so.

AC - The relatively high error is due to the fact that the algorithm in its original form (feedback with integrator and proportional term) is not constructed to deal with sudden changes, such as starting strong SAI in 2080. Rough "dry-testing" experiments with a simple box model emulator for GMST suggest that this can not be resolved by simple re-tuning. Therefore we have made an ad-hoc improvement by resetting the integrated error term, as described in the manuscript, which reduces but does not eliminate the "cooling overshoot" observed in fig. 2b. We will state this more explicitly in the revised manuscript.

AC - We have stated that tuning does not solve the problem completely in L187-188.

RC1 - Figure 2: I found the panels confusing, in that you're mixing and matching units.

AC - We understand the confusion. To improve the figure we will add separate y-axes in blue for the temperature errors. This will avoid the mixing of units.

AC - The temperature y-axis has been moved to the right in Fig. 3.

RC1 - Line 257: Per the above comment, maybe change "well" to "adequately".

AC - Yes, we will change this.

AC - Changed to "adequately" in L208.

RC1 - Line 292: This is correct but also a strawman argument. You didn't try to restore the climate completely.

AC - Thank you for pointing this out. We will rephrase this more factually such that we still acknowledge local differences without pretending it to be our goal to restore the climate completely.

AC - We have removed the strawman argument in L234-235.

RC1 - Figure 5: I'm having trouble making sense of how important these results are. I wonder if you could compute z-scores (or something like that) so I would know whether the CAM minus WACCM differences are large compared to the natural variability of WACCM.

AC - Good point. We will add hatching to the revised figure 5 in the manuscript to indicate significance. Additionally, hatching will be applied to other figures that we think will benefit from this.

AC - 95% significance tests have been performed for all shaded contour data in the main body. Stippling is added to Figs. 5-7. Besides, a new figure shows how large model differences of (potential) temperature (WACCM-CAM SAI-REF) are w.r.t. the model mean SAI-CNT (appendix Fig. A4).

RC1 - I did not find the paragraph on lines 376, nor Appendix B, terribly convincing. If you heat the stratosphere by 24°C, you are going to have substantial influences on ozone, and we know that ozone has an influence on surface climate. I would be more comfortable if you simply said that this is what you did, its effects of ozone changes on climate are likely smaller than the effects of the stratospheric heating on climate, and this should be explored further. That puts you on much safer ground.

AC - We initially included this section to provide some more background into the matter, but this indeed poses a risk as the processes involved are more complex than shown here, requiring further

analysis. As our article is meant to convey a useful protocol for practitioners, it would indeed be safer to state the things you mention. We will do so in a revised version of the manuscript.

AC - We have tried to present the appendix section more as a results section, and less as an argument for what we did. Meanwhile, we also found that Kravitz et al., 2019 seemed to have some similar thoughts on ozone in WACCM, and we used this to back up our motivation in the main text. The changes are made to L298-309 and appendix B.

RC1 - Lines 400-402: Kravitz et al. (2014) demonstrated that the controller is likely robust to these sorts of differences. That gives some confidence that your controller indeed can handle this.

AC - That is a good suggestion. We will mention this in the revised text.

AC - We mention this in L346-347.

RC1 - Lines 427ff: See recent work from the Cornell group, specifically led by Farley or Brody. They're doing the initial steps of what you propose.

AC - We will read their relevant publications and refer to these in the outlook.

AC - We added this as part of a possible path for future development in L369-371.

RC1 - Code availability: Journals tend to want a fixed repository (e.g., Zenodo) rather than a changeable repository (Github).

AC - We will make a Zenodo repository for the revised version.

AC - The Zenodo is mentioned in L391.

---

## Author Comment (AC8)

*This manuscript by de Jong at al. is trying to do two things: first, it is trying to describe how to prescribe stratospheric aerosols forcing in a different model. Second, it's trying to show the surface climate differences in two different CESM1 versions (CESM2-CAM6, CESM1 and CESM2-WACCM6) forced by the same aerosol fields (prescribed, in the case of CAM).*

*The main thing I take issue with, in the first part, is that essentially what they are proposing to do is not novel at all. Prescribing an aerosol field externally is something that has been done before multiple times not just for volcanism (the entirety of CMIP6 historical simulations used prescribed aerosols) but for SRM, too. Conveniently, the authors leave out all the references of this, trying to pass it off as "novel". Personally, I deeply dislike this way of not acknowledging past work (or to avoid performing a literature search that would have made past works emerge) to hype up one's own.*

*Here are three main examples:*

- *Tilmes et al. (2016) proposed a prescribed aerosol forcing file for models with no interactive sulfate cycle. Such forcing field was then used in Xia et al. (2017) and in one of the models for G6sulfur (CNRM) as described in Visioni et al. (2021).*

- *Still in G6sulfur, the model MIP prescribed their own aerosol field and scaled them up in their two fully-coupled versions, also as described in Visioni et al. (2021).*

- *Finally, Tilmes et al. (2024) proposed a new experiment for CCMI with a new aerosol field for SAI, also describing climatic differences in WACCM when fully interactive vs prescribed aerosols are used, with much more details provided on how to prescribe the aerosols fields in different models provided in Jörimann et al. (2025). Reading Jörimann et al. (2025) could also illuminate for the authors how hard it is to prescribe aerosol fields in other models' version, as one needs to get optical properties that might be treated differently in different models with different ways to translate aerosol size distribution to forcing.*

*None of this is ever acknowledged in the (rather short and non comprehensive in general) cited literature, giving the reader the impression that this is the first time something like this has been attempted (cue the word "novel" used frequently in the text, and also in the Key Points).*

*So, what's novel? The use of a scaled-up, mono-dimensional control algorithm (not that different from what was used in multiple G6 models, see above) used in multiple publications starting in MacMartin et al. (2014)? That's hardly new – and the simplified controller used here hardly seem well tuned, considering the pretty large errors shown in Fig. 2!*

**AC** – We thank the reviewer for the extensive comments and literature suggestions. However, we suspect there was a misunderstanding about the scope of our work, and in particular, what aspects thereof we consider to be "novel". At no point has it been our intention to pretend that we came up with the idea to force models with aerosol fields, either directly or scaled similarly as in previous work, and it never occurred to us that our text might be read in this way. Reviewer 1 seems to confirm that this is a reasonable assumption.

Instead, our paper intends to present and test a useful protocol for a specific application of forcing climate models with aerosol fields, namely, mimicking GLENS / ARISE-like SAI simulations in situations where running a full stratospheric chemistry model is impractical, for example due to computational constraints. For this specific application, our method is novel to our best knowledge, and modellers from several groups expressed interest in using it.

The submitted manuscript aims to disseminate our protocol and outline potential avenues for future use, which fits within the stated purposes of GMD ("development and technical papers, describing

developments such as new parameterizations or *technical aspects of running models*" and "model experiment descriptions, including experimental details and project protocols".

We will take care to clarify our intentions. We will pay particular attention to reformulating the abstract, which we agree can be read as if the scope of our work is broader than it is. Please find further specific changes we intend to apply to the manuscript below.

**Relation of our protocol with previous work**

From preliminary tests we found that simply inserting WACCM-derived aerosol fields in CAM will lead to rather different temperature outcomes (i.e., an aerosol field taken from WACCM cannot stabilise GMST in CAM). Such a forcing can be scaled manually, but this can be cumbersome, especially when running non-standardized SAI forcing scenarios. Hence, for our application of mimicking WACCM forcing in CAM we explored the possibility of using a feedback controller combined with a new method of scaling. Our protocol builds on two prior strands of work:

- SRM simulations with feedback controllers. For us, SAI simulations with a full-fledged chemistry model are most relevant, although feedback controllers have also been applied on simpler simulations and are currently being applied to Marine Cloud Brightening. Examples involving SAI include the simulations with CESM(WACCM) for G6sulfur (Visioni et al. 2021), GLENS (Kravitz et al., 2017), and an inclusion of the same feedback controller in the ARISE-SAI protocol (Richter et al., 2022).

- Prescribing (stratospheric) aerosol forcing to models that do not simulate them explicitly; for us, applications in SAI are the most relevant, but others exist (volcanic eruptions). Examples include the CNRM and MPI-ESM models that participated in G6sulfur (Visioni et al., 2021), which may be forced by the aerosol forcing set provided by Tilmes et al., 2015 (CNRM) or a similar set produced by the same model (MPI-ESM) and scaled by means of choice, CESM(CAM-chem) (Xia et al., 2017), and an inclusion of using prescribed forcing data in an SAI protocol for CCMI (Tilmes et al., 2024). Regarding this protocol, the REMAPv1 code (Jörimann et al., 2024) is proposed as a viable method to calculate aerosol properties if the input dataset does not contain the fields required by the model.

Our method adds to the existing literature in two ways:

- Most importantly: Combining a feedback controller with aerosol forcing

- Allowing for non-linear relationships between AOD and other aerosol variables

**Combining a feedback controller with aerosol forcing.** In the literature mentioned above and suggested by the reviewer, we have not encountered a publication documenting the combined use of a feedback controller and prescribing aerosols.
Tilmes et al. (2015) provides a predetermined forcing scenario, which is directly prescribed to the model in Xia et al. (2017). Both CNRM and the two versions of MPI-ESM in Visioni et al. (2021) scale such prescribed forcing by multiplying the AOD with some year-dependent factor to achieve the temperature target. Automatising the feedback controller - as has been done in e.g. GLENS and ARISE with full aerosol models - obviously can save considerable time, especially when exploring new scenarios in the simpler model setup (as out SAI2080 in CESM-CAM).
In short, one original contribution of our paper is simply to have worked out a practical way of combining these previous strands of work in order to allow SAI simulations in the (computationally cheap) CAM to mimic SAI simulations in WACCM as closely as possible in terms of (tropospheric / near-surface) climate impacts, here using GMST as target variable - and to document it sufficiently clearly

(including some practical details such as the iteration to get a sufficiently good feedforward) that other groups can hopefully save themselves time and errors if they wish to implement something similar.

**Nonlinear relationship between AOD and other variables.** Apart from using a feedback controller for scaling aerosol forcing - more precisely, AOD - we also use non-linear fits to relate other variables such as aerosol mass, size and surface area to AOD. As far as we can see from the literature, this was not the case in previous studies where scaled aerosol forcing was used. A consequence of this approach is that aerosol field properties may not change when scaling the forcing strength. Tilmes et al. (2022) for example find that a linearly increasing forcing strength for CNRM in G6sulfur resulted in a linearly increasing SAD because of the fixed aerosol size, which is most likely dependent on the forcing strength (Niemeier et al. 2011, Niemeier and Timmreck 2015). Though the relationship between AOD and the dominant mode aerosol mass is fairly linear in our simulations, other variables show different relations (fig. 1 of the manuscript).

**Translating aerosol fields for use in different models** The prescription of aerosols into models that require different aerosol properties than provided requires the use of Mie calculations as done by the REMAPv1 algorithm developed by Jörimann et al. (2025). In our analyses, we did not need such translations as WACCM output may be directly prescribed to CAM. Our method can therefore be applied by anyone using a compatible pair of models, regarding the stratospheric forcing, but not when different input is required. In our revised version of the manuscript, we will stress that we have not validated the case for non-matching input fields, e.g. with a less general description in the abstract. It might be possible to combine the REMAPv1 algorithm with our method if one needs to translate between incompatible pairs of models (i.e., the model receiving forced aerosols requires different inputs than the model generating the forcing). We will mention this in the discussion.

Summarizing, the proposed method uses an improved scaling not seen before in prescribed-aerosol SAI modelling and can easily be combined with existing feedback controllers. This makes it a valuable tool and source of inspiration for modelling groups that will perform their own SAI experiments (focussed on impact).

We will add the above discussion on prior works and how our work relates to them to the revised manuscript.

AC – In response to the above suggestions, we have made the following revisions to the manuscript:

- Throughout the manuscript, we have revised our language to avoid the term "novel method", and instead refer to our approach as a "novel application of ..." (L4), to more accurately reflect the nature of our contribution.

- In the abstract and discussion, we now clearly specify that our study focuses exclusively on WACCM-CAM, i.e. a model combination which shares similar variable sets and therefore does not require variable conversion (L3-6, L35-36, L378-387).

- Additionally, we have clarified in the abstract and introduction that our method is specifically intended for research on tropospheric and/or (sub)surface impacts of SAI, to appropriately set expectations for its scope (L9-11, L36-37).

- We have incorporated the discussion of relevant previous work above into the introduction to provide a clearer context for our study (L38-65).

- The discussion now includes a reference to the recent work on aerosol translation by Jörimann et al. (2025), which we believe strengthens the applicability of our method (L382-387).

*The second part then offers a description of surface climate differences between CAM and WACCM. This part (which would also not be particularly fitting for GMD on its own) is awfully lacking as well.*

*First, I don't see any kind of discussion of the biases between the two models' versions without SAI, and I think that would be a fundamental starting point to understand the sources of differences to a forcing. Even very similar versions of CESM2-WACCM6 that differ by the inclusion or not of tropospheric chemistry can present relevant distinctions in some modeled trends (see i.e. Davis et al., 2023), let alone two basically different models. Aside from surface climate, most of the differences in the stratospheric response are also left very vague (for instance, Section 3.3 essentially doesn't offer anything more than guesses ("These features occur in Control as well and are assumed to be model differences") that don't really provide much confidence to the reader that the authors know what they're talking about. The entire paragraph from l. 376 to l. 384 is very high level, does not cite any of the relevant literature about the impacts of stratospheric ozone changes on surface climate (to begin with, but the list is very long, see Bednarz et al., 2022).*

*I do not wish to claim that novelty in itself makes a study worthy of being published, and actually I think quite the contrary: but a study that wishes to claim novelty as its main strength, fails to deliver and also ignores relevant literature that would help frame this work in the broader context (out of curiosity, the manuscript has 24 references, of which 7 are in the first paragraph to prove/disprove the permissibility of SRM research or to discuss climate change, one of them is a non-peer reviewed link to CarbonBrief, and one a never accepted preprint from 2023, casually including one of the authors of this piece) is setting itself up to fail for anyone who has any familiarity with the field.*

**Literature review - additional points**

We thank the referee for pointing out the Bednarz et al., 2022 paper, which can serve as a useful comparison case.

We would like to keep the initial, more general remarks about the controversies around SRM, because we believe that even an otherwise technical paper should provide a minimum of context regarding SRM's controversies and dilemmas. The Futerman et al., 2023 paper is accepted and we expect to have the link to the final version by the time this manuscript is published. The Pflüger et al., 2024 reference is very relevant because the same forcing method is used and is correctly referring to a published peer-reviewed paper.

**Validation**

We now address concerns raised by referee 2 regarding the quality of our validation. What we present is a tool that essentially emulates the stratospheric aerosol forcing in a model that does not have stratospheric chemistry. The purpose of its validation is to show how well the tropospheric climate responds to this applied forcing, so that modellers may decide to use our proposed method or not. For our validation we used the following metrics (in decreasing order of importance:

- Can the method achieve the set climate target (GMST)?
  GMST can be stabilised to the desired level, which is the main objective of our protocol. This works nicely when reproducing WACCM scenarios (in our case, SAI2020). When generating new scenarios which strongly differ from the WACCM scenarios used as input (in our case, SAI2080), discrepancies occur but these are limited and explainable and can be reduced further by fine-tuning the procedure (in our case, resetting the integrator term of the feedback) .

- Does the method reproduce the response to SAI of the tropospheric climate?
  In particular we verify if (surface) temperature, precipitation and wind responses to SAI in CAM resemble responses in WACCM for equivalent scenarios. The model discrepancies we find are not

larger than inter-model difference in the absence of SAI. Inter-model differences are evaluated in Reference and RCP8.5. Please note that the scope of our validation is merely to check whether our method introduces large additional biases (relative to existing inter-model differences in the absence of SAI), not to provide an in-depth comparison between CAM and WACCM more broadly.

- Does the CAM simulation successfully control all degrees of freedom targetted in WACCM, even if not included in CAM's feedback controller? Does this depend on changes in the scenarios (e.g. SAI2080 instead of SAI2020) or model version (e.g. switching to higher resolution)?
  We find that even if we only control for GMST, other targets used in the unerlying wACCM simulations (T1, T2) are fairly well reproduced if CAM is used to generate a similar scenario (SAI2020). The same does not hold for the rather extreme SAI2080 scenario. This discrepancy can be explained by the long-term AMOC changes incurred during the prolonged warming under SAI2080. As we also discuss in the manuscript, this discrepancy could likely be mitigated by expanding our protocol with additional degrees of freedom, i.e. including several aerosol patterns associated with different injection latitudes in WACCM. We agree with the reviewer that it would be interesting to test this. However, for the time being, we are restricted to using for the validation the simulations we have now (which were partly developed for other pieces of analysis), therefore we cannot do more than discussing this (current) limitation and suggesting an expansion to multiple aerosol patterns as an avenue for future development.

- Stratosphere:
  Our method should not be used by anyone looking for high fidelity in the stratosphere, as mentioned in our abstract and introduction. For such research interest, clearly using WACCM is a more viable approach. Our interpretations on the differences in the stratosphere such as in section 3.3 are considered to be a nice-to-have and do not comprise an in-depth analysis of the mechanisms resulting in the observed patterns of change. They are a simple check whether the limited representation of stratospheric processes - such as atmosphere-ozone interaction or other things that could affect stratospheric heating - should be expected to have impacts in tropospheric circulation (as for example suspected by McCusker 2015)

Regarding the above, we believe that the validation in the current version of the manuscript is fit for purpose. We agree that we should avoid potential misunderstandings about the range of problems out methods should, or shouldn't, be applied to. In the revised version of the manuscript, we will state more clearly that our method is not to be used for prescribing aerosols in SAI simulations where high fidelity in the stratosphere is required, but rather as a tool for simulations meant to analyse the tropospheric/oceanic impacts of geoengineering.

**AC** – Response to the above concerns:

- We have now cited Bednarz et al., 2022 to acknowledge the potential implications of prescribing ozone using a fixed scenario, particularly with regard to its influence on the tropospheric climate (L306-307).

- We have retained the broader discussion and acknowledged the ongoing controversies surrounding SRM, as we believe this context is important for situating our work.

- The Futerman et al., 2023 preprint has since been published and is now cited as Futerman et al., 2025. Any other preprints, including the suggested Jörimann et al., 2025, are now clearly marked as "in review" in the manuscript (L19).

- The suggested analysis of model biases lies beyond the scope of the paper because our focus is on repeatedly achieving the above-mentioned goals. However, we have made additional efforts throughout the text to clearly emphasize that our study is intended to support research on

tropospheric and surface/oceanic impacts of SAI only (as mentioned previously). Additionally, we now show how close the temperature response in CAM is to that in WACCM in terms of the change in SAI-Control (appendix Fig. A4).

*This manuscript, in another venue but GMD, could provide some interesting analyses of what happens when you prescribe the same aerosol field in different model's versions given much more in depth, careful analyses, exploring the sensitivities, performing further experiments (i.e. different scenarios, which are available in CESM2-WACCM6, could be compared with the same prescribed CAM6 results to understand if the differences observed are a function of magnitude, of the specifics of the AOD pattern, of the broader atmospheric response, of the underlying emission scenario; the ozone field could have actually been changed; CAM6 simulations with fixed SSTs could have been performed to understand the actual forcing response; different patterns of AOD from different injection locations could have been combined to understand the linearity of the response, etc. The authors do acknowledge that "It might be worthwhile to expand the method for additional use cases." But, given the paucity of interesting results otherwise, I think they should be the ones, and not somebody else at a later date). In its current form, however, I cannot recommend publication in this venue.*

**On the additional suggested experiments**

A long list of additional simulations and analyses is suggested by referee 2 to develop a deeper understanding of the stratospheric forcing response. We have no doubt that many of these would greatly contribute to the community's understanding of stratospheric aerosol processes and aerosol-induced forcing. However, the scope of this paper is not a detailed research aerosol processes related to SAI or provide an in-depth model comparison, but to present a useful protocol for practitioners interested in (surface) climate impacts. Therefore the suggested fixed-SST experiment, while very interesting for wider research on aerosol forcing, are beyond the scope of this study.

We agree that some other suggested experiments would be nice to add here. For example, as regards additional scenarios, it would be nice, for the purpose of additional validation, to try using CESM-CAM aerosol fields based on a particular CESM-WACCM simulation (e.g. what we call SAI2020 here) to reproduce another existing CESM-WACCM simulation (e.g., ARISE-like scenarios (Richter et al., 2022)). Also, as discussed above, it would be nice to expand our method by using aerosol patterns from several injection latitudes to test for linearity and improve the performance of new scenarios (here SAI2080) on additional degrees of freedom. Unfortunately, we have to make do with simulations originally designed for other research projects. Adding the suggested new simulations would require a very substantial amount of additional computational and human-time effort, which we cannot commit and which do not weigh up against the gains achievable for this paper. We therefore do not intend to perform the suggested additional experiments.

AC – We have not run new simulations for the reasons provided above. We provided an overview of the simulations we did run in Table 1.

We hope the mentioned revisions address the reviewer's concerns and improve the clarity and framing of our work.

**References**

Bednarz, E. M., Visioni, D., Banerjee, A., Braesicke, P., Kravitz, B., & MacMartin, D. G.: The overlooked role of the stratosphere under a solar constant reduction. Geophysical Research Letters, 49, e2022GL098773. `https://doi.org/10.1029/2022GL098773`, 2022.

Davis, N. A., Visioni, D., Garcia, R. R., Kinnison, D. E., Marsh, D. R., Mills, M., et al.: Climate, variability, and climate sensitivity of "Middle Atmosphere" chemistry configurations of the Community Earth System Model Version 2, Whole Atmosphere Community Climate Model Version 6 (CESM2(WACCM6)). Journal of Advances in Modeling Earth Systems, 15, e2022MS003579. `https://doi.org/10.1029/2022MS003579`, 2023.

Jörimann, A., Sukhodolov, T., Luo, B., Chiodo, G., Mann, G., and Peter, T.: A REtrieval Method for optical and physical Aerosol Properties in the stratosphere (REMAPv1), EGUsphere [preprint], `https://doi.org/10.5194/egusphere-2025-145`, 2025.

Kravitz, B., Robock, A., Tilmes, S., Boucher, O., English, J. M., Irvine, P. J., Jones, A., Lawrence, M. G., MacCracken, M., Muri, H., Moore, J. C., Niemeier, U., Phipps, S. J., Sillmann, J., Storelvmo, T., Wang, H., and Watanabe, S.: The Geoengineering Model Intercomparison Project Phase 6 (GeoMIP6): simulation design and preliminary results, Geosci. Model Dev., 8, 3379–3392, `https://doi.org/10.5194/gmd-8-3379-2015`, 2015.

Kravitz, B., MacMartin, D. G., Mills, M. J., Richter, J. H., Tilmes, S., Lamarque, J.-F.,... Vitt, F.: First simulations of designing stratospheric sulfate aerosol geoengineering to meet multiple simultaneous climate objectives. Journal of Geophysical Research: Atmospheres, 122, 12,616–12,634. `https://doi.org/10.1002/2017JD026874`, 2017.

McCusker, K. E., Battisti D. S., and Bitz C. M.: Inability of stratospheric sulfate aerosol injections to preserve the West Antarctic Ice Sheet. Geophys. Res. Lett., 42, 4989–4997 `https://doi.org/10.1002/2015GL064314`, 2015.

Niemeier, U., Schmidt, H. and Timmreck, C.: The dependency of geoengineered sulfate aerosol on the emission strategy. Atmosph. Sci. Lett., 12: 189-194. `https://doi.org/10.1002/asl.304`, 2011.

Niemeier, U. and Timmreck, C.: What is the limit of climate engineering by stratospheric injection of SO2?, Atmos. Chem. Phys., 15, 9129–9141, `https://doi.org/10.5194/acp-15-9129-2015`, 2015.

Richter, J. H., Visioni, D., MacMartin, D. G., Bailey, D. A., Rosenbloom, N., Dobbins, B., Lee, W. R., Tye, M., and Lamarque, J.-F.: Assessing Responses and Impacts of Solar climate intervention on the Earth system with stratospheric aerosol injection (ARISE-SAI): protocol and initial results from the first simulations, Geosci. Model Dev., 15, 8221–8243, `https://doi.org/10.5194/gmd-15-8221-2022`, 2022.

Tilmes, S., Mills, M. J., Niemeier, U., Schmidt, H., Robock, A., Kravitz, B., Lamarque, J.-F., Pitari, G., and English, J. M.: A new Geoengineering Model Intercomparison Project (GeoMIP) experiment designed for climate and chemistry models, Geosci. Model Dev., 8, 43–49, `https://doi.org/10.5194/gmd-8-43-2015`, 2015.

Tilmes, S., Visioni, D., Jones, A., Haywood, J., Séférian, R., Nabat, P., Boucher, O., Bednarz, E. M., and Niemeier, U.: Stratospheric ozone response to sulfate aerosol and solar dimming climate interventions based on the G6 Geoengineering Model Intercomparison Project (GeoMIP) simulations, Atmos. Chem. Phys., 22, 4557–4579, https://doi.org/10.5194/acp-22-4557-2022, 2022.

Tilmes, S., Bednarz, E. M., Jörimann, A., Visioni, D., Kinnison, D. E., Chiodo, G., and Plummer, D.: Stratospheric Aerosol Intervention Experiment for the Chemistry-Climate Model Intercomparison Project, EGUsphere [preprint], `https://doi.org/10.5194/egusphere-2024-3586`, 2024.

Visioni, D., MacMartin, D. G., Kravitz, B., Boucher, O., Jones, A., Lurton, T., Martine, M., Mills,

M. J., Nabat, P., Niemeier, U., Séférian, R., and Tilmes, S.: Identifying the sources of uncertainty in climate model simulations of solar radiation modification with the G6sulfur and G6solar Geoengineering Model Intercomparison Project (GeoMIP) simulations, Atmos. Chem. Phys., 21, 10039–10063, `https://doi.org/10.5194/acp-21-10039-2021`, 2021.

Xia, L., Nowack, P. J., Tilmes, S., and Robock, A.: Impacts of stratospheric sulfate geoengineering on tropospheric ozone, Atmos. Chem. Phys., 17, 11913–11928, `https://doi.org/10.5194/acp-17-11913-2017`, 2017.

---

## Author Response (AR2)

**Response to editor comments**

Thank you for submitting the revised manuscript which addresses most of the reviewer comments. Regarding the scope and novelty, my decision is that it is sufficient for publication. However, the report #2 by one of the reviewers has several important minor revision suggestions that I agree will be important to implement. Please address these and we can proceed with publication:

AC - We are glad to hear your decision. Here, we respond to the reviewer's concerns. Line numbers correspond to the revised pdf version, whereas line numbers in blue correspond to the latexdiff file.

- EC Title wise, the title doesn't respect the guidelines set out by GMD itself. As a "Development and technical paper", the guidelines state "If the model development relates to a single model then the model name and the version number must be included in the title of the paper. If the main intention of an article is to make a general (i.e. model independent) statement about the usefulness of a new development, but the usefulness is shown with the help of one specific model, the model name and version number must be stated in the title. The title could have a form such as, "Title outlining amazing generic advance: a case study with Model XXX (version Y)"." Therefore, a more correct title for this paper would be "A computationally efficient method to model Stratospheric Aerosol Injection using prescribed aerosols in a lower complexity version of the same model: a case study using CESM(CAM) and CESM(WACCM)" which fairly represents what the paper tries to describe.
- AC Thank you for pointing this out. We indeed only show results for CAM-WACCM, requiring a change of title. Your suggestion was very helpful and has been used for the new title, with a minor addition of being able to run both similar and alternate SAI experiments.
- EC Again, I find the beginning of the Introduction not very professional for a scientific paper. The first context-setting paper for climate change is a blog post when at least you could cite the WMO State of the Climate report for 2024, which has a ISBN and a Permalink and has been peer-reviewed (see https://library.wmo.int/records/item/69455-state-of-the-global-climate-2024, https://library.wmo.int/idur1/4/69455) two opinion pieces pro and against SRM research and continues being vague with qualitative statements ("Arguably", "Seems to combine", "highly controvertial"...).
  AC Yes this is a fair point. We have limited this paragraph to introducing SRM and SAI in a more neutral way, i.e. without touching upon the ongoing debate around its potential usage or its effectiveness (L15-21,L16-27). The blog post is now replaced by the WMO State of the Climate report (L16,L18).
- EC As another note, the Feder et al. (204) paper that is cited hasn't been accepted for publication, it's not In Review anymore.
- AC We have removed the Feder et al., 2024 reference (L29–30,L36), but it is hard to find a substitute. However, this proves the point of having very few studies with high-res SAI simulations. Also, the Farley et al. (2025) has not yet been accepted and replaced by a similar older paper: Farley et al. (2024) (L463,512). The Jörimann et al. (2025) paper has been accepted and changed accordingly (L474,L527).
- EC L. 35: why not just say "a method to approximate" rather than the awkward and inprecise "(approximately) model"
- AC That sounds more accurate indeed. See changes in L33,L39.
- EC L. 216: "rough dry-testing" is not a scientific term, nor is it something that would mean anything to a reader.
- AC We performed some idealized tests with the feedforward-feedback controller in which GMST was determined using the FAIR climate model emulator (Millar et al., 2017) and random noise. We mention this in L193-195, L202-205.
- EC L. 263-265: aside from this phrase being needlessly convoluted, it is also way too generic. Using

the current scenario of injections (excluding higher-latitude injections), in WACCM almost no land areas have residual temperature changes "on the order of a few degrees". At most you can see areas where the residual is below 1C (absolute), and this can be better quantified, especially in the differences between CAM and WACCM (that might be due to the fact that the injection pattern in WACCM might not be the same in CAM to reduce those residuals).

AC - Yes we agree and have removed this phrase. To improve the quality of this section, we now indicate more clearly about what differences (e.g. inter-model or SAI minus Reference) we are talking, use more accurate numbers (e.g. typical inter-model differences much less than 1°C), noted a few thoughts on the warming bias and describe the seasonal variations a bit more (L246–264,L257–280).

EC - L. 309: why is potential temperature included, and what is the significance of that compared to air temperatures for the LS? Potential temperatures is less related to the heating rates increases due to sulfate absorption, and normally just air temperature changes are analyzed in other works looking at stratospheric response (see the Bednarz paper cited after). So if the authors want to shot PT rather than air temperature, they should justify why.

AC - Potential temperature has been included as it makes thermal wind balance calculations in pressure coordinates easier. However, it is true that plotting temperature is the more sensible thing to do here. This has been changed in Figs. 7,A5 and the accompanying text (L309,311,313) (L325,327,329).

EC - L. 334: the CFC dependence has been analyzed much more in more recent works, such as Tilmes et al. (2021)

AC - Yes this is a great resource. Tilmes et al. (2021) suggest that the combined effect of available chlorine concentrations and aerosol size lead to a peak in ozone decrease per injected amount in the initial decade after starting SAI, leading to limited further ozone decrease in the subsequent decades. We figured the section about ozone is better suited for the Discussion and moved most of it there. We refer to this work in L411 L460.

EC - The new Discussion/Outlook section is also left entirely vague ("a vastly different forcing scenario", "generally much closer", "qualitatively similar" etc.). Please quantify some of the outcomes of your paper. Due to its length (and to its lack of clarity), it would also be useful to separate the Discussions vs Conclusions sections explicitly.

AC - We agree that some of the outcomes can be quantified better. On top of that, we think the aims of the paper could be stated more clearly. We made improvements by 1) quantifying some of the changes in T0,T1,T2 (new Table 4 in Results) and 2) relating inter-model changes of key variables to their interannual variability through a performance index (new Figure 8 in Results, introduced in section 2.4 in Methods). These indicators of performance are mentioned in the aims of the paper (L71–77) (L78–84), displayed in Results (sections 3.1.1 and 3.4 (mind numbering error in latexdiff)), used in the Discussion (L350–361,L378–379,L383–387,L400–404) (L380–392,L416–418,L427–430,L450–453). We have also structured the previous Discussion/Outlook section in a Discussion (with a part on validation results and a part on tested use cases), Conclusions and Outlook.

**References**

Farley, J., MacMartin, D. G., Visioni, D., and Kravitz, B. (2024): Emulating inconsistencies in stratospheric aerosol injection, Environmental Research: Climate, 3, 035 012, https://doi.org/10.1088/2752-5295/ad519c

Jörimann, A., Sukhodolov, T., Luo, B., Chiodo, G., Mann, G., and Peter, T. (2025): REtrieval Method for optical and physical Aerosol Properties in the stratosphere (REMAPv1), Geoscientific Model Development, 18, 6023–6041, https://doi.org/10.5194/gmd-18-6023-2025

Millar, R. J., Nicholls, Z. R., Friedlingstein, P., and Allen, M. R., (2017). A modified impulse-response representation of the global near-surface air temperature and atmospheric concentration response to carbon dioxide emissions. Atmospheric Chemistry and Physics, 17, 7213–7228, https://doi.org/10.5194/acp-17-7213-2017

Tilmes, S., Richter, J. H., Kravitz, B., MacMartin, D. G., Glanville, A. S., Visioni, D., (2021). Sensitivity of total column ozone to stratospheric sulfur injection strategies. Geophysical Research Letters, 48, e2021GL094058, https://doi.org/10.1029/2021GL094058